# Transdermal electroosmotic flow generated by a porous microneedle array patch

Shinya Kusama[1], Kaito Sato[1], Yuuya Matsui[1], Natsumi Kimura[1], Hiroya Abe [1], Shotaro Yoshida[1] & Matsuhiko Nishizawa [1,2✉]

A microneedle array is an attractive option for a minimally invasive means to break through the skin barrier for efficient transdermal drug delivery. Here, we report the applications of solid polymer-based ion-conductive porous microneedles (PMN) containing interconnected micropores for improving iontophoresis, which is a technique of enhancing transdermal molecular transport by a direct current through the skin. The PMN modified with a charged hydrogel brings three innovative advantages in iontophoresis at once: (1) lowering the transdermal resistance by low-invasive puncture of the highly resistive stratum corneum, (2) transporting of larger molecules through the interconnected micropores, and (3) generating electroosmotic flow (EOF). In particular, the PMN-generated EOF greatly enhances the transdermal molecular penetration or extraction, similarly to the flow induced by external pressure. The enhanced efficiencies of the EOF-assisted delivery of a model drug (dextran) and of the extraction of glucose are demonstrated using a pig skin sample. Furthermore, the powering of the PMN-based transdermal EOF system by a built-in enzymatic biobattery (fructose / $O_2$ battery) is also demonstrated as a possible totally organic iontophoresis patch.

[1] Department of Finemechanics, Graduate School of Engineering, Tohoku University, Sendai, Japan. [2] Division for the Establishment of Frontier Sciences of the Organization for Advanced Studies, Tohoku University, Sendai, Japan. ✉email: nishizawa@tohoku.ac.jp

Telemedicine with home healthcare systems has attracted attention as an essential technology for addressing the growing problems of an aging society and for improving medical care during disasters/infectious epidemics[1,2]. Skin patches for transdermal diagnosis/treatment are typical devices for home healthcare, and there has been progress in improving skin compatibility, multifunctionality, and disposability[3–5]. Iontophoresis has been studied to accelerate transdermal penetration/extraction by typically applying a continuous low voltage current[6,7]. The iontophoretic transdermal drug delivery systems have been actively studied[8–13], and are already commercialized for the fast dosing of drugs for dermal anesthesia (LidoSiteTM, Vyteris Inc.)[8], post-operative pain relief (IonsysTM, Alza)[9], and anti-migraine (ZecuityTM, NuPathe Inc.)[10]. Recently, a totally organic transdermal drug delivery patch containing a built-in enzymatic battery has been reported[11]. Also, the transdermal iontophoretic extraction of interstitial fluid (ISF) (a process sometimes called 'reverse iontophoresis') has been gaining attention as a sample collection method for medical diagnosis[14–17], including continuous glucose monitoring with the GlucoWatch® system[18]. The mechanism of iontophoretic transport consists of the electro-osmotic flow (EOF) of the solvent (water) as well as the electrophoresis of the charged molecules themselves. EOF is generated by the preferential movement of mobile cations (or anions) in the fluid conduits containing fixed anions (or cations) (Supplementary Fig. 1). Under physiological conditions, the skin acts as a cation-selective matrix (isoelectric point: ~4.5)[19], and therefore an EOF can be generated in the anode-to-cathode direction (i.e., in the same direction as migration of cations)[6,7]. The technical difficulty of these transdermal iontophoresis applications arises from the barrier functions of the stratum corneum, the outermost layer of skin (~20 μm thickness)[7,17]. Since iontophoresis is a DC technique, the electrical barrier of the stratum corneum (resistance, ~10 MΩ)[20,21] makes it difficult to induce stable transdermal currents. Also, because of the barrier function to mass transfer, only small molecules (< ca. 500 Da) can be a candidate for transdermal delivery and collection[22,23]. Taken together, the issues to be addressed for advanced transdermal iontophoresis are (1) lowering the transdermal resistance, (2) transporting of larger molecules, and (3) generating a larger EOF.

A microneedle array is an attractive option for a minimally invasive means to break through the skin barrier for drug delivery (Supplementary Table 1)[24,25]. Needles of a microscale length (usually <1 mm) make it possible to pass the stratum corneum without reaching blood vessels and nerves[24,25]. The drug molecules including hormones and vaccines can be delivered by coating on the conventional solid needles[26,27], or by incorporating within needles made of dissolvable polymers[28–31]. The swellable needles have also been used for extraction of skin ISF for diagnostic analysis[24,32–34]. For the purpose iontophoresis, the microneedles should contain fluid conduits like the cylindrical hole of traditional injection needles[35–37], but the mass fabrication of a hollow structure in a μm-scale needle has been found to be difficult[37]. Another type of fluid permeable microneedle is the solid-based porous microneedle (PMN)[38–45]. We have recently realized a mechanically stable ion-conductive PMN containing a large volume of interconnected pores[44,45]. However, EOF generation via the PMN has not been studied so far.

In this study, we succeeded in the generation of a larger transdermal EOF by using PMN, aiming at applications for efficient drug delivery (penetration) and analysis of ISF (extraction), as illustrated in Fig. 1a. The ion-conductive PMN significantly lowered the transdermal resistance by partial breaking of the stratum corneum, and its modification with a hydrogel containing sulfonic groups realized the generation of the transdermal EOF in the anode-to-cathode direction. The PMN-generated EOF

enhanced the transdermal molecular penetration or extraction, similarly to the flow induced by external pressure. We employed modification of the negatively charged hydrogel according to the polarity of the skin tissue. The higher the density of sulfonic groups in the hydrogel, the larger was the flow velocity of the transdermal EOF. The EOF-assisted delivery of a model drug (fluorescently labeled dextran, 10,000 Da) and the extraction of glucose was demonstrated using a pig skin sample. The driving of the transdermal EOF system with an enzymatic biobattery (fructose/$O_2$ biobattery) was also demonstrated to explore the possible construction of a totally organic EOF patch (Fig. 1b).

## Results

**Preparation and characterization of porous microneedles (PMN).** Figure 2a shows a photograph of the array of 37 porous microneedles (PMN) at 1 mm intervals on a porous substrate (φ 8 mm, 0.5 mm thickness), all of which are made of poly-glycidyl methacrylate (PGMA). The PMN is opaque due to light scattering in the microporous structure. The length of the PMN itself was designed to be 250 μm so that it can penetrate to the epidermic layer through the stratum corneum without reaching the sensory nerves in the dermis layer of skin[24,25,46], and thus the subjects did not feel any discomfort from its insertion. Each PMN (28° angle of φ0.13 mm cone, 0.25 mm height) protrudes from a cylindrical supporting post (300 μm height, 450 μm diameter) for reliable penetration into the skin (detail design, Supplementary Fig. 2). The supporting posts induce local stretching of the skin and facilitate effective insertion of the microneedles (probability: >90%) as examined by OCT imaging[45]. Needle density was optimized by examining the insertion probability; the higher density of needle decreased the probability because of the lack of local stretching of skin. Mechanical stability of the porous microneedle (compression fracture force: ca. 2.5 N) was confirmed (Supplementary Fig. 3), and the cytocompatibility was checked by the Live-Dead test of dermal fibroblast cultured with the chip (Supplementary Fig. 4). The SEM image indicates that the PMN is made up of aggregated particles of ca. 0.5 μm in diameter with interconnecting pores of ca. 1.0 μm diameter on average. This observation is roughly consistent with the quantitative estimation of the internal structure of the PMN by a surface area measurement based on Brunauer-Emmett-Teller (BET) theory[47]; the slope and the intercept of the linear region of the BET plot for the adsorption of nitrogen gas (Fig. 2b) give the specific surface area of 7.23 m²/g, which corresponds to the particle size of 0.65 μm, as explained in "Methods".

For the measurement of DC ionic resistance of the PMN itself, a PMN chip was set on a highly conductive agarose hydrogel containing Ringer's solution and connected to a source-meter via agar salt bridges to minimize contact resistance, as shown in Fig. 2c–i. The resistance of a PMN chip was found to be ~3 kΩ, indicating that roughly 25% of the pores form continuous channels through the PMN, as calculated from the porosity (ca. 50%) of PMN and the resistivity of the Ringer's solution loaded in the PMN (ca. 61 Ω cm)[48]. The intact skins of subjects' arms showed significantly larger and scattered resistance (0.2–5 MΩ) as shown in Fig. 2c–ii due to the insulating nature of the stratum corneum, the outermost layer of skin. The obtained values are roughly consistent with those reported previously[20,21] despite the resistance value of intact skin being dependent on the method of measurement, humidity, and the measured part of body. Importantly, by using the PMN, the transdermal resistance was lowered and stabilized to 40–150 kΩ (Fig. 2c-iii), which is of great advantage, ensuring stable, safe generation of the transdermal EOF. Figure 2d shows the equivalent circuit of the DC conduction system of PMN and skin[49]. The large resistance of the stratum

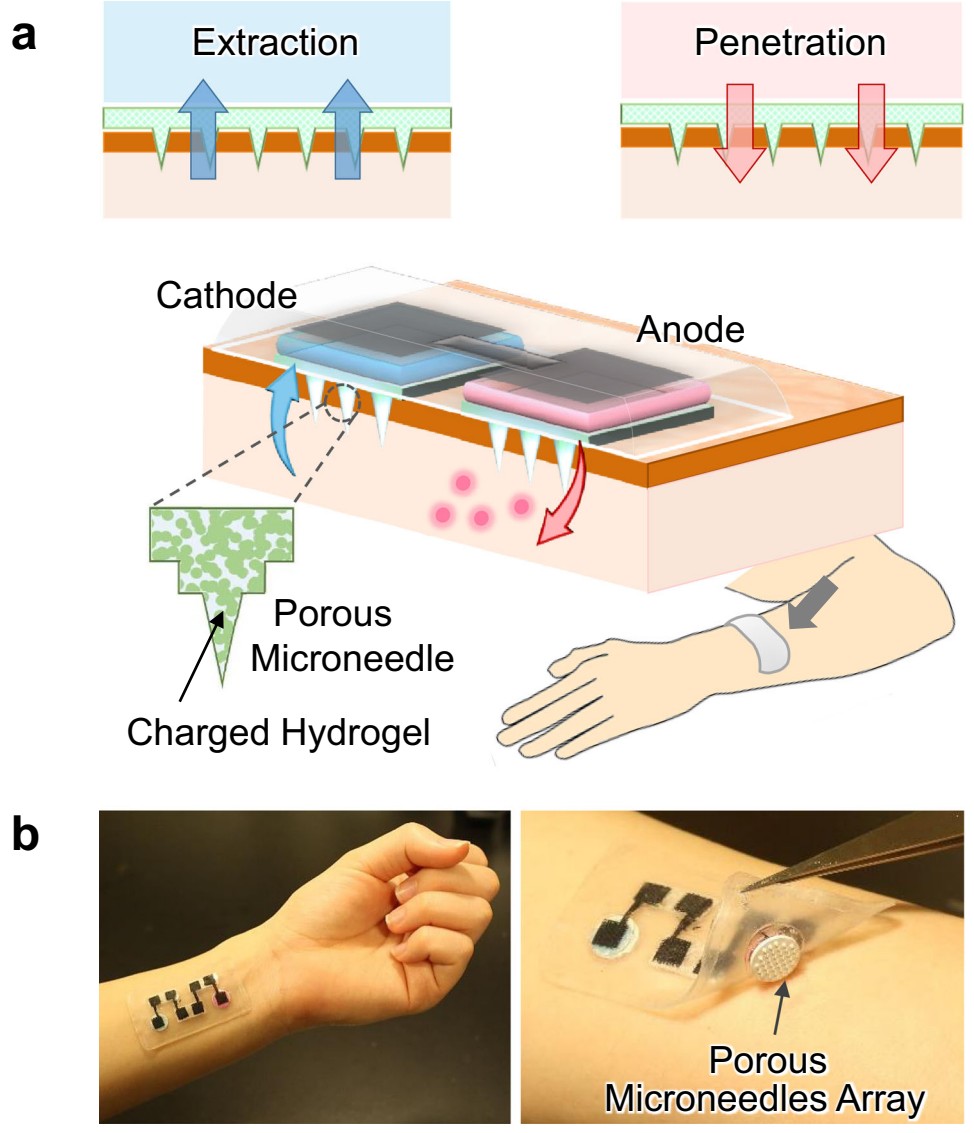

**Fig. 1 Transdermal electroosmosis by a porous microneedle array patch. a** Conceptual illustration of the electroosmotic penetration/extraction for efficient drug delivery and analysis of ISF. The flow direction from anode to cathode is the case when the porous microneedles are modified with a negatively charged hydrogel. **b** Photographs depict a structural overview of the combination of the biobattery (four cells connected in series) and the porous microneedle.

corneum in the MΩ range determines the net resistance of the intact skin. On the other hand, the stabilized lower net resistance with PMN mainly reflects the resistance of the epidermis and dermis.

The preforming of holes in the stratum corneum by inserting conventional (not porous) microneedles has been studied for effective transdermal fluid transport. For example, the Prausnitz group has recently reported an optimized microneedle patch combined with absorbing materials for collection of ISF[50]. Li et al. reported the enhanced permeation of drugs into the skin by pretreatment with a solid polymer microneedle[51]. Although the preforming of holes is simple and practical, the holes in the stratum corneum are not stable[52,53] because of the physical closure due to the elasticity of skin tissue. Figure 2e shows the DC resistance of the arm skin of two subjects monitored during the placement and removal of a PMN. The intact resistance of skin before the placement of PMN has a large variation between subjects (difference of 600 kΩ in the present case) that can be decreased to <100 kΩ by inserting PMN. By removing the PMN after ca. 40 min, the skin resistance was suddenly increased

reflecting the physical closure of the pores, followed by physiological wound healing[52,53]. These results indicate that preforming holes with microneedles has limitations in the stable and efficient iontophoresis through skin, while PMN is a unique tool that can maintain a lower resistance and smaller variation between subjects, which should be of great advantage for transdermal iontophoresis with superior reproducibility.

**Electroosmotic flow (EOF) through PMN modified with charged hydrogel.** By using the side-by-side Franz cell[6,54] shown in Fig. 3a, the efficiency of EOF generation (EOF strength) was evaluated to study the effect of the modification of poly-2-acry-lamido-2-methylpropane sulfonate (PAMPS) into the porous structures of PMN. We employed the modification of negatively charged hydrogel according to the polarity of epidermis, while it is also possible to select modification of positively charged hydrogel to generate FOF of opposite direction (Supplementary Fig. 5). DC current densities of 0.25, 0.5, and 1.0 mA/cm$^2$ were applied, and the flow velocity of EOF was calculated from the

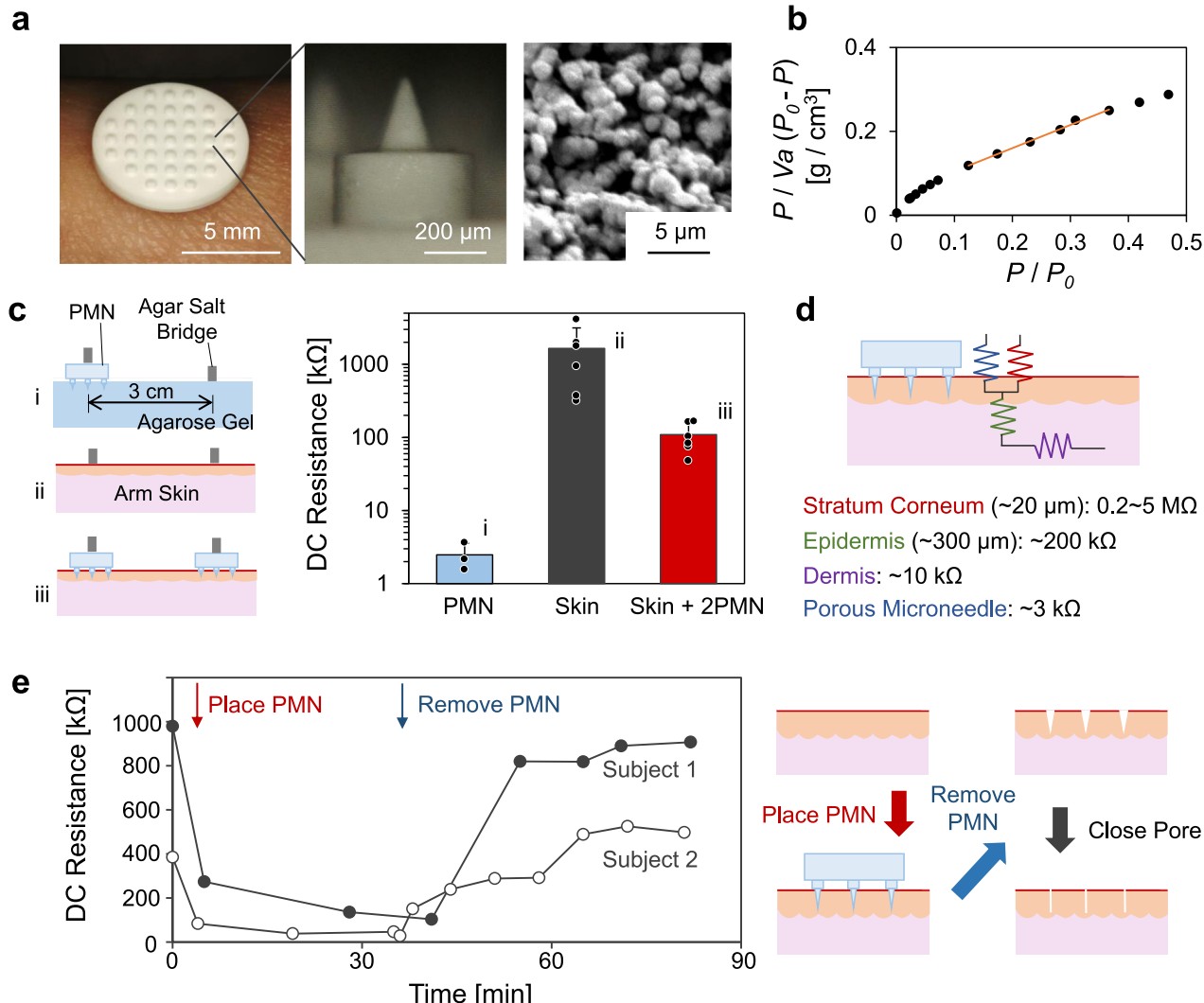

**Fig. 2 Structural and electrical characterization of porous microneedles (PMN). a** Representative photographs of a chip of PMN arrays and a representative cross-sectional SEM image of PMN ($N = 3$ independent samples). **b** A BET plot for nitrogen adsorption in PMN. $V_\alpha$ [cm³/g] is the adsorbed gas volume per the weight of PMN at STP. $P$ and $P_0$ are the monitored and the saturation pressure [Pa], respectively. **c** The DC resistance of (i) a PMN ($N = 3$ independent experiments; mean ± standard deviation (SD)), (ii) human arm skin ($N = 6$ independent experiments for 3 human subjects; mean ± SD), and (iii) the skin with PMNs ($N = 6$ independent experiments for 3 human subjects; mean ± SD), measured by applying 1 μA. **d** Schematic of the equivalent DC circuit of a layered structure of a skin with the experimental resistances of each layer. **e** The time-course of DC resistance of two human subjects' arm skin, in which the red arrow and the blue arrow indicate the timing when the PMN was placed and removed, respectively. During the period of PMN insertion, a tubular salt-bridge was put on the PMN as in (**c**-iii). Right illustration draws the pore closure after the removal of PMN.

movement of the water surface in a horizontal capillary (inner diameter, 1.4 mm) of the cell. Figure 3b shows the flux of water transport through the naked PMN and that modified by PAMPS. The PAMPS-filled PMN generated a water flux proportional to the current density, while the water flux through the naked PMN was negligibly small because the functional groups of PGMA (glycidyl group and diol group) have no charge at around neutral pH conditions. The fixed negative charge of sulfonic group of PAMPS should be the origin of the EOF according to the following equation[55].

$$U_{eo} = K_{eo} I \qquad (1)$$

$$K_{eo} = -\frac{\varepsilon \zeta \rho}{\eta} \qquad (2)$$

where $U_{eo}$ [μL/(cm²·h)] is the electroosmotic flow velocity, $I$ [mA/cm²] is the current density applied to the fluid conduit, $\varepsilon$ [F/m] is the dielectric constant of the solvent (water), $\zeta$ [mV] is the zeta

potential of the negative charge fixed in the conduit media, $\rho$ [Ω·cm] is the specific resistance the sample, and $\eta$ [Pa·s] is the viscosity of the solvent (water). The value of $K_{eo}$ [μL/(mA·h)] is the ratio of $U_{eo}$ and $I$, and can be regarded as a standardized indicator of the EOF strength of the conduit. Figure 3c shows the variation of $K_{eo}$ with the concentration of monomer (AMPS) used for the modification of PAMPS. As expected, the denser PAMPS produced by a higher concentration of the monomer tends to show larger $K_{eo}$ (larger EOF strength) mainly due to the increase in the effective $\zeta$ in Eq. (2). Importantly, for the dilute concentrations AMPS (0.05 and 0.1 M), $K_{eo}$ shows larger values deviating from the linear trend. From the SEM images in Fig. 3d, it was found that the porous structure of PMN still can be recognized even after the polymerization of 0.05 M AMPS, while the PAMPS made from 0.5 M AMPS looks to fully fill the pores. Figure 3e illustrates speculation from the results of Fig. 3c, d, where the amount of PAMPS made from 0.05 M AMPS is not enough to fill the pores and results in hollow microchannels with

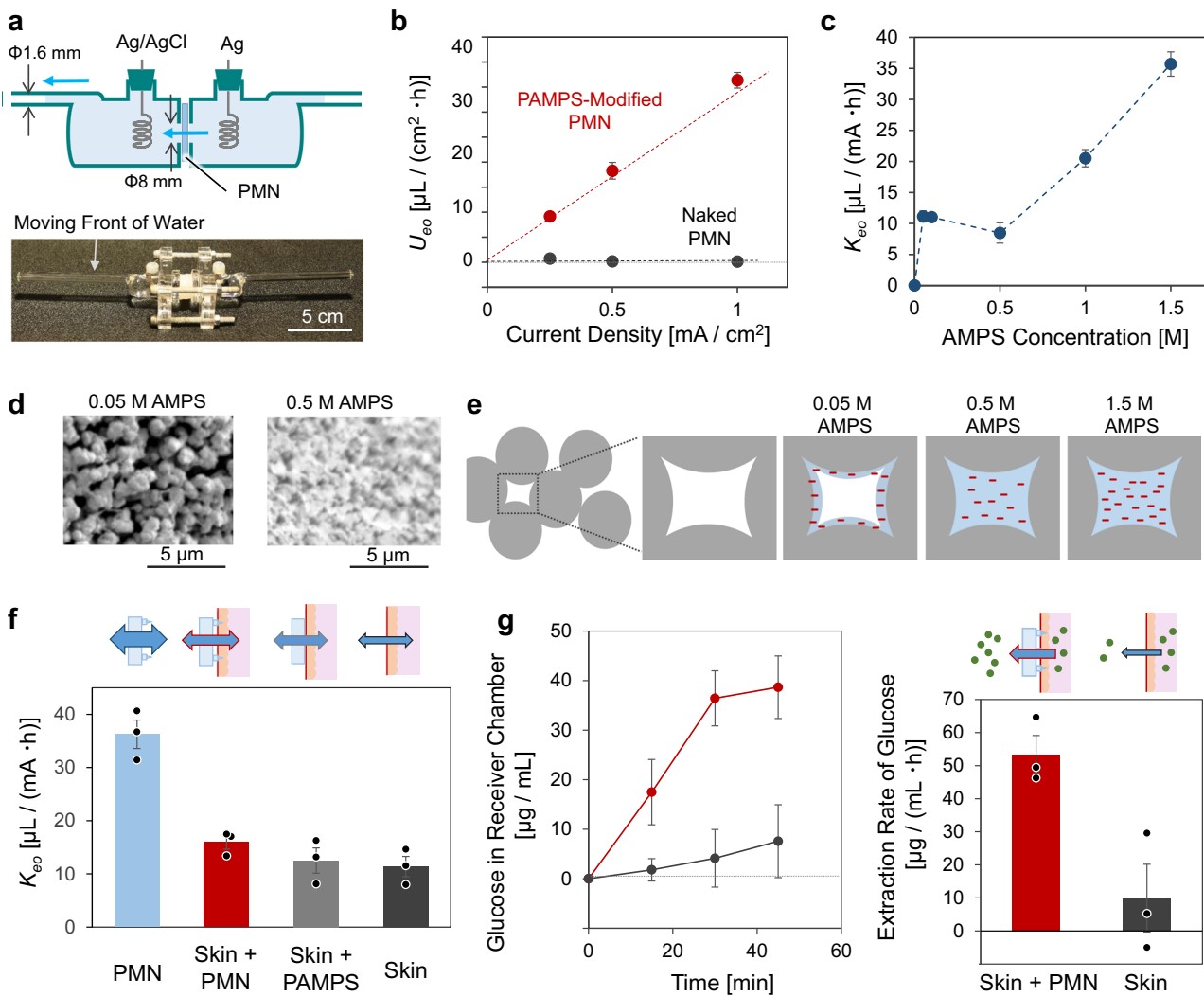

**Fig. 3 Transport of water and small molecules. a** A side-by-side Franz cell for measurement of the flux of water transport through a PMN under DC current application. **b** The flow velocity of EOF as a function of current density applied for a naked PMN (black) and a PAMPS-filled PMN prepared from 1.5 M AMPS (red) ($N = 3$ independent experiments; mean ± SD). **c** The plot of $K_{eo}$ (slope of $U_{eo}$ vs. current density) against the AMPS concentration used for the modification of PAMPS ($N = 3$ independent experiments; mean ± SD). **d** Representative SEM images of cross sections of PMNs modified with PAMPS made from 0.05 and 0.5 M AMPS ($N = 3$ independent experiments). **e** Schematic illustration of pores in the PMN modified with PAMPS from various concentrations of AMPS. **f** The EOF strength ($K_{eo}$) through a PAMPS-filled PMN made from 1.5 M AMPS (blue), a pig skin with the PMN (red), a pig skin with the PAMPS-filled porous PGMA plate (gray bar) and an intact skin (black) ($N = 3$ independent experiments; mean ± SD). **g** The time-course of glucose concentration in receiver chamber (plot) and the rate of extraction of glucose (bar) from the intact skin (black) and the skin with the PAMPS-filled PMN (red) ($N = 3$ independent experiments; mean ± SD). pH 7 McIlvaine buffer was used.

fixed charges on their inner walls that is a suitable condition to effectively generate EOF[56]. On the other hand, although the PAMPS filling the pores (>0.5 M AMPS) would impede the flow of water to some extent, the larger amount of fixed charge (1.0 and 1.5 M AMPS) can cause larger amounts of EOF generation. A higher concentration of AMPS than 1.5 M sometimes caused cracks in the needle due to larger degree of swelling rate during its polymerization. In the following sections, the PMN filled with a dense PAMPS (1.5 M AMPS) was used for the demonstration of the transdermal water transport with small molecules, while the hollow PMN with PAMPS (0.05 M AMPS) was used for the transport of larger molecules.

**Transdermal transport of water and small molecules**. The red bar in Fig. 3f depicts the EOF strength ($K_{eo}$) of the transdermal water transport through the PAMPS-filled PMN (1.5 M AMPS) measured using the Franz cell and a piece of pig skin (~4-mm

thick) (Supplementary Fig. 6). The transdermal EOF with the PMN shows a $K_{eo}$ value between those of the PMN alone (blue bar) and the pig skin (black bar). The control experiment with the PAMPS-filled porous PGMA plate without microneedles (gray bar) showed an intermediate $K_{eo}$ value, indicating the significance of the partial breaking of the stratum corneum with the needles. The negative charge of mucopolysaccharides and proteins (e.g., keratin) in epidermal tissue is known to serve as the ionic conduit to generate EOF in the experiments with skin alone[6,7]. The additional transdermal EOF was produced by the synergy of the negative charge of PAMPS and the breaking of the stratum corneum with the needles; the PAMPS-filled PMN promoted the transdermal water flux ca. 2 times (red bar against black bar). Figure 3g shows the results of extraction of glucose (5.6 mM in donor chamber) through an intact pig skin (black plot and bar) and the skin with PAMPS-filled PMN (red plot and bar) at the transdermal current of 2.0 mA/cm². The concentration of 5.6 mM

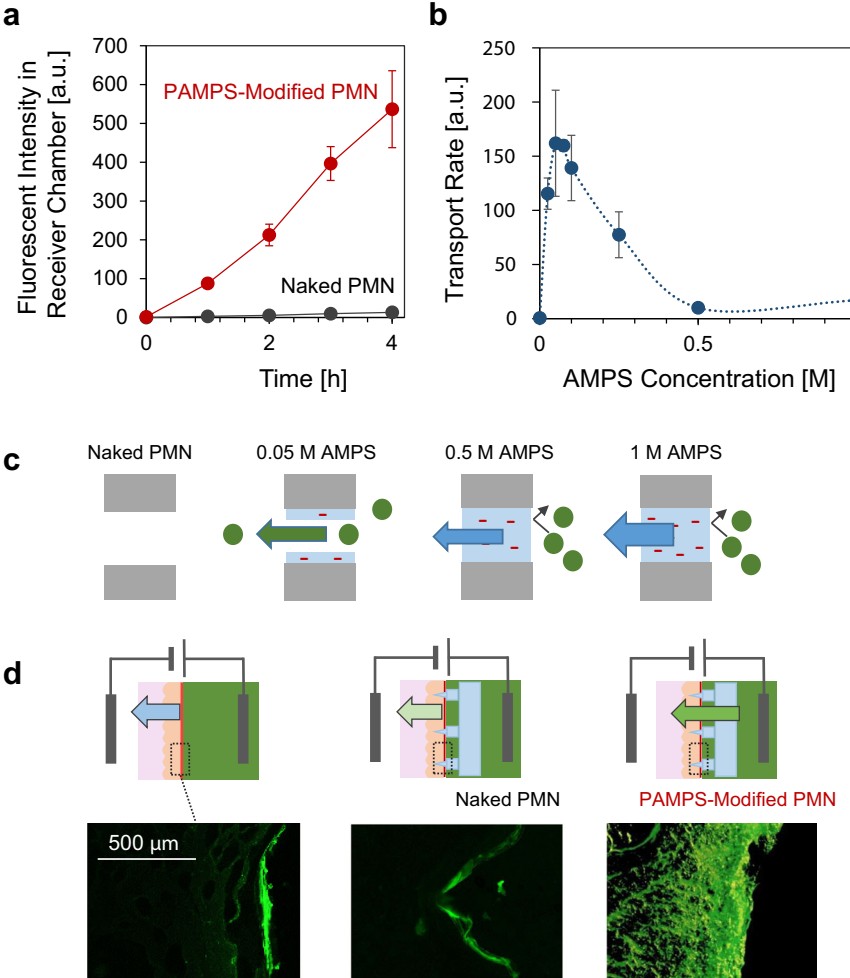

**Fig. 4 Transport of large molecules. a** The time-course of fluorescent intensity of FITC-dextran transported to the receiver chamber through the naked PMN (black) and the PAMPS-modified PMN (red) at 3 mA/cm² ($N = 3$ independent experiments; mean ± SD). **b** The transport rate of FITC-dextran at 3 mA/cm² as a function of AMPS concentration for modification of PAMPS ($N = 3$ independent experiments; mean ± SD). **c** Schematic illustration of the sieving effect of the pore-filling PAMPS for larger molecules. **d** Cross-sectional fluorescence images of pig skins after the transdermal application of 0.5 mA/cm² for 30 min without a PMN (left), with a naked PMN (center) and with a PAMPS-modified PMN (0.05 M AMPS) (right) ($N = 3$ independent experiments for each sample).

is typical of blood glucose that is known to correlate to the concentration of glucose in ISF, while the change of concentration has some time lag[57,58]. The concentration of glucose in the receiver side was analyzed every 15 min using a glucose assay kit[59]. As expected, the use of the PAMPS-filled PMN increased the extraction speed of glucose (molecular weight, 180 Da). It is worth noting that the promotion effect was higher for the transport of glucose (ca. 5 times) than that of water flow (ca. 2 times), reflecting the higher barrier function of stratum corneum for glucose than water. Therefore, the breaking of stratum corneum by the PAMPS-filled PMN would be of advantage for sampling the solution of which glucose concentration is relatively close to the ISF. The concentration of glucose in ISF has been proved to reflect the blood glucose level[15,17], and therefore the low-invasive sampling of the ISF with the PAMPS-filled PMN could be important in daily personal diabetic treatments, although a prior calibration still be required[17,60]. Also, the EOF-based transdermal administration of chemicals of small size (lidocaine[9], ascorbic acid[61,62], nicotine[63] etc.) could be promoted by using the PAMPS-filled PMN.

**Transport of large molecules through PMN.** Figure 4a shows the EOF-assisted transport of a fluorescently labeled model drug

(FITC-dextran, ca. 10,000 Da) through the PMNs at 3.0 mA/cm². The chemical structure of FITC-dextran is shown in Supplementary Fig. 7[64], and its size is comparable to that of many drugs, including insulin (ca. 6000 Da). The solution in the receiver side was sampled every hour, and the fluorescent intensity was measured with a spectrophotometer. The naked PMN without fixed charge did not produce EOF, and thus the transport of dextrin was not observed. On the other hand, the accelerated transport of dextran to the receiver side was confirmed for the PMN modified with PAMPS made from 0.05 M AMPS. These experiments were conducted at pH 6 in order to see clearly the EOF-assisted transport of FITC-dextran (pKa ≈ 6.4), where the proportion of mono-anionic FITC-dextran is higher than di-anionic (Supplementary Fig. 7). It is worth noting that the transport direction at pH 7 was the same as pH 6 due to the strong EOF enough to transport even the di-anionic FITC-dextran against the electrophoresis of anionic species (Supplementary Fig. 8). Figure 4b shows the variation of the transfer rate of dextran against the concentration of AMPS used for the modification of PAMPS. In contrast to the positive correlation in the water transport (Fig. 3c), the transport of dextran (a large molecule) showed a maximum peak at around 0.05 M of AMPS, and almost no transport was occurred in the range of higher concentration of

monomers (0.5 and 1 M). We believe that the above results can be understood by the illustrations in Fig. 4c. In the case of the naked PMN, the molecular movement was almost negligible because of the absence of fixed charge that is necessary for EOF generation. At the AMPS concentration of around 0.05 M, the transport of dextran effectively occurs with EOF through the negatively charged hollow microchannel in the PMN. Finally, at the higher concentration of AMPS than 0.5 M, the transport of dextran is hindered by the sieving effect of the pore-filling PAMPS, while the flow of solvent (water) was accelerated (Fig. 3).

**Transdermal dosing of large molecules.** The transdermal dosing of a model drug (FITC-dextran) into a piece of pig skin was demonstrated by using the PAMPS-modified PMN (0.05 M). By setting the pig skin so that its stratum corneum faced the donor side, 0.5 mA/cm² was applied for 30 min, followed by washing and observation of frozen sections with a fluorescence microscope (Fig. 4d). In the absence of PMN (left), only the stratum corneum was stained by the fluorescent dextran due to its poor permeability to stratum corneum. When a naked PMN is inserted into the skin (center), the penetration of the dextran was still limited at the surface area of the skin. In contrast, the deeper penetration of dextran was clearly seen for the skin with the PAMPS-modified PMN (right), indicating that the modification of PAMPS was necessary to generate a strong enough EOF to effectively promote the administration of dextran. Unfortunately, it was hard to fix the diffusion state of molecules in the dermis in the sliced sample for microscope observation. In order to monitor the penetration progress, an in situ observation system with a multiphoton microscope is under development.

**Transdermal EOF driven by enzymatic biobattery.** A user-friendly iontophoresis skin patch requires the integration of a lightweight and safe power source to drive the transdermal electroosmosis. Enzymatic biobatteries[65–69], which utilize enzymes as an electrocatalysis for anodic oxidation of biochemical fuels like sugars and cathodic reduction of atmospheric O₂, can be an option for such an organic disposable power source for an iontophoretic transdermal patch[11]. Figure 5a illustrates the combination of the PMNs and a fructose/O₂ biobattery, which consists of a series-connected array of four couples of anode and cathode electrodes in order to generate sufficient transdermal EOF by the boosted voltage[65]. Figure 5b depicts the typical performance of the biobattery composed of the fructose dehydrogenase (FDH)-modified flexible carbon fabric (CF) anodes (1.0 cm²), and the bilirubin oxidase (BOD)-modified CF cathodes (1.0 cm²), taken on a cotton cloth containing 0.2 M D-fructose. It was found that the open-circuit voltage was roughly four times that of a single fructose/O₂ biobattery (ca. 0.7 V), and the maximum output current reached 0.52 mA/cm². Figure 5c shows the typical time-course of the transdermal current measured on a pig skin, whose resistance was lowered to be ca. 10 kΩ by using the PMNs. The observed current of around 0.2 mA/cm² will generate EOF effective for the transdermal iontophoresis (Fig. 3). Figure 5d demonstrates the promoted dosing of FITC-dextran to a pig skin through the PAMPS-modified PMN (0.05 M AMPS). Cotton with 0.3 mL McIlvaine buffer containing 0.2 M D-fructose and 0.75 mg/mL FITC-dextran was set between the FDH-modified anode and the PMN as schematically illustrated. The cross-sectional fluorescence image, taken after a 1 h application of the power of the biobattery, indicates the deep penetration of dextran induced by the EOF through the PMN. Finally, the extraction of glucose from the pig skin presoaked in a 5.6 mM glucose solution was demonstrated in Fig. 5e by using the PAMPS-filled PMN (1.5 M AMPS). The 40 μL portion of the

solution in the cotton receiver between the PMN and the BOD-modified cathode was analyzed by a glucose assay kit after the 1 h application of the power of the biobattery. The transport of a significant amount of glucose was found (red bar), whereas a control experiment without the biobattery (black bar) indicated that the passive extraction of glucose was almost negligible. The combination of the present sampling technique with an advanced organic sensor patch[68–70] will realize a totally organic patch for the analysis of interstitial fluid.

## Discussion

In this study, three significant functions of the originally developed ion-conductive porous microneedle (PMN) were demonstrated aiming at applications for iontophoresis: (1) the lowering and stabilization of skin resistance by low-invasive partial breaking of the stratum corneum, (2) the molecular permeability through the wholly interconnected micropores, and (3) the generation of large electroosmotic flow (EOF) by charge-modification in the micropores. By utilizing these advantages of the novel charge-modified PMN, we succeeded in demonstrating the controlled generation of transdermal EOF for the first time. The PMN-generated EOF would play the role similar to the flow induced by external pressure, and accelerate progress in the development of home healthcare patches for transdermal drug delivery and the diagnosis of interstitial fluid.

The physicochemical barrier function of human skin is known to allow penetration of only small molecules less than about 500 Da[22,23], and has limited the drug candidates for transdermal dosing. The PMN in this work was made up of aggregated particles of ca. 0.5 μm with interconnecting pores of ca. 1.0 μm, as seen by both microscope observation and BET measurement. This dimension of the microchannels in the PMN is large enough to transport most kinds of larger drugs including insulin (ca. 6000 Da) and vaccines. In order to further increase the EOF strength of the charge-modified PMN, increasing the porosity would be an effective approach. However, we found that a sufficient mechanical strength of the PMN for penetrating skin (compression fracture force, ca. 2.5 N) was ensured by limiting the porosity to <50%. The replacement of the present epoxy resin (PGMA) material of the PMN by a biodegradable polymer such as poly (lactic-co-glycolic acid) (PLGA) is the necessary future study in order to guarantee biocompatibility even upon breaking it in skin. Since the preparation method for porous PGMA using porogen was not applicable for the pre-polymerized PLGA, we have been developing a novel process to prepare porous microneedles of PLGA, and will report it in near future.

The modification of the hydrogel PAMPS containing negatively charged sulfonic groups showed a significantly enhanced efficiency in producing EOF. The density of PAMPS can be optimized depending on the size of target molecules. For water and small molecules like glucose, the higher rate of EOF-assisted transport was observed for denser PAMPS (higher density of negative charge) fully filling the micropores. On the other hand, for the larger molecules, the optimum amount of PAMPS modification was better when reduced so as to form negatively charged hollow microchannels in the PMN. There are many papers of animal experiments, in which iontophoretically dosed larger molecules (peptide, enzyme, hormone, vaccine etc.) exerted their function in vivo[71–73]. The activity of glucose oxidase (GOx) was determined to be maintained after the application of the electroosmotic flow (Supplementary Fig. 9).

Owing to the lowered skin resistance obtained by the use of PMN, a series-connected enzymatic biobattery (fructose/O₂ battery) can serve as the integrated power source to drive the EOF-assisted transdermal iontophoresis. Although we used fructose

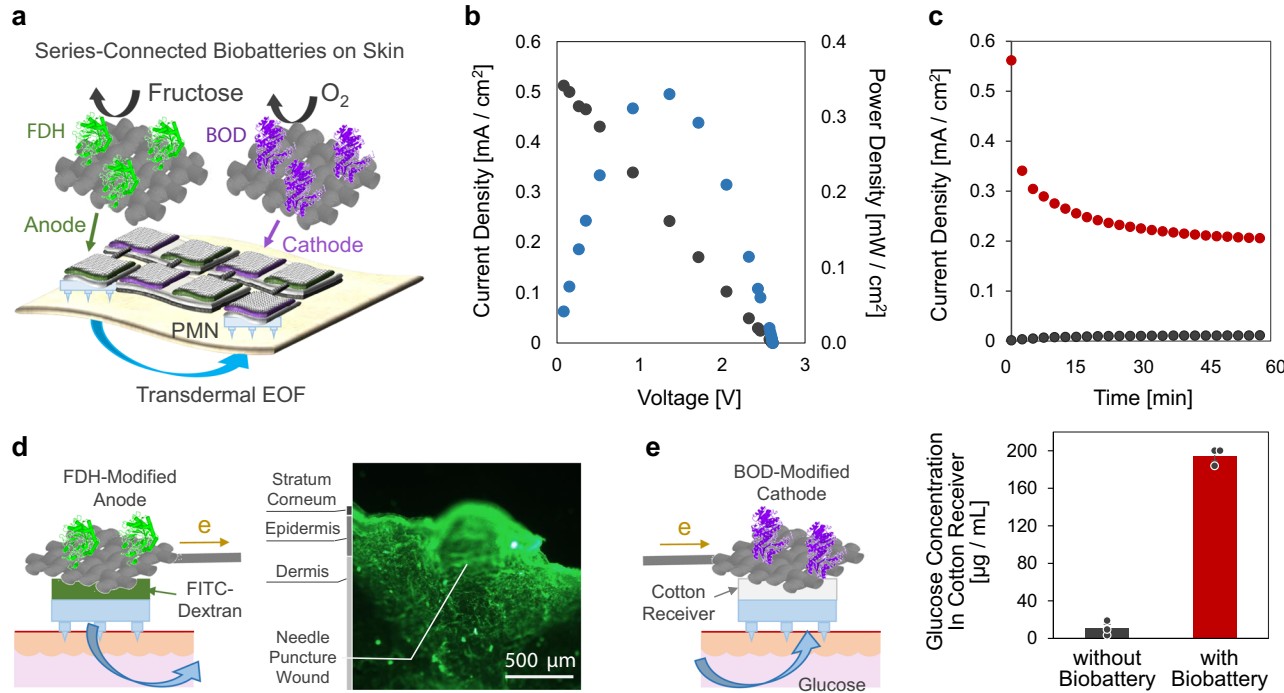

**Fig. 5 Integration of enzymatic biobatteries. a** Illustration of series-connected enzymatic biobatteries (fructose/O$_2$ biobattery) for driving transdermal EOF through the PMN. **b** The current–voltage and power–voltage characteristics of the biobatteries on a cotton cloth containing physiological saline and 0.2 M D-fructose. **c** The typical time-course of the transdermal current generated through a piece of pig skin by the biobattery with (red) and without (black) PMN. **d** Illustration of the FDH-anode on a pig skin combined with a PAMPS-modified PMN (0.05 M AMPS) and cotton containing 0.2 M fructose and 0.75 mg/mL FITC-dextran. The representative cross-sectional fluorescent image was taken after a 1 h application of the power of the biobattery ($N = 3$ independent experiments for each sample). **e** Illustration of the BOD-cathode on a pig skin combined with a PAMPS-filled PMN (1.5 M AMPS) and a plain cotton receiver. The concentration of glucose extracted with (red bar) and without (black bar) the application of power of biobattery for 1 h ($N = 3$ independent experiments; mean ± SD).

because its electrooxidation can be directly catalyzed by FDH without any electron mediator, other fuel molecules such as glucose and lactic acid can also be used for driving the skin patches in the presence of suitable mediators[66–69]. The successful demonstrations shown here using a built-in biobattery proves the future possibility of a totally organic EOF-based skin patch that is safe and truly disposable.

## Methods

**Fabrication of porous microneedle array.** The array chip of naked porous microneedles (PMN) made of poly-glycidyl methacrylate (PGMA, FUIJFILM Wako Pure Chemical) was prepared by the combination of a molding process and the porogen method[44]. Briefly, a female mold for a PMN was made with PDMS by following the previously reported protocol of two-step molding. A plate of poly (methyl methacrylate) was drilled by a 5-axis CNC machine (PRODIA M45, Modia Systems) with a square end mill (diameter 300 μm, Nissin tool) and a conical drill bit (tip diameter < 10 μm, tip taper angle 13.5°, custom-made by DoCraft). The PMNs were made by using two stock solutions. A monomer stock solution was prepared by mixing the monomer glycidyl methacrylate (FUIJFILM Wako Pure Chemical, 10 mL), crosslinker trimethylolpropane trimethacrylate (Sigma-Aldrich, 5.23 mL), and crosslinker triethylene glycol dimethacrylate (FUIJFILM Wako Pure Chemical, 15.7 mL). A porogen stock solution was prepared by dissolving 4.0 g polyethylene glycol (10 kDa, FUIJFILM Wako Pure Chemical) in 20 g of 2-methoxyethanol at 50 °C. Just after mixing the monomer and porogen stock solutions (6:7 in volume), the photoinitiator Irgacure 184 was added (1 wt% to the monomer), followed by pouring the mixture into the PDMS mold. The mold with the mixture was put under vacuum at ca. $2.0 \times 10^2$ Pa for 45 min to ensure that bubbles were removed and all the cavities of the mold were filled with the solution. Photopolymerization was conducted by irradiation of 365 nm UV light for 1 h to make a microneedle array chip composed of 37 needles with an interval of 1 mm on a planar substrate (ϕ8 mm × 0.5 mm in thickness). By dissolving the porogen with 60 °C ethanol/water (1:1 volume) for 12 h, the naked PMN was obtained. The mechanical compression fracture force of a tip of the naked PMN was 2.5 N ± 0.26 (Supplementary Fig. S3). The porosity of the naked PMN was estimated as ca.

50 vol% as follows,

$$(\text{Porosity}) = \frac{V_{\text{pore}}}{W_{\text{dry}}/d + V_{\text{pore}}} \quad (3)$$

where $W_{\text{dry}}$ [g] is the weight of the dry PMN, $d$ [g/cm$^3$] is the density of solid PGMA without pore, and $V_{\text{pore}}$ [cm$^3$] is the pore volume of the PMN. The pore volume was calculated from the difference of weight of PMN in dry and wet conditions (pure water). The density of solid without pores was calculated by a method using Archimedes's principle[74]; the PMN was immersed in a cylindrical test tube containing pure water, and the volume was calculated by the difference of height of the water.

For the modification of PAMPS in pores of PMN, the naked PMN chips were immersed in a 10 mL aqueous solution containing 0–1.5 M of AMPS (FUIJFILM Wako Pure Chemical), 0.10 M of MBAAm (FUIJFILM Wako Pure Chemical), 100 μL of ammonium persulfate (APS; an initiator; 10 wt% in water, FUIJFILM Wako Pure Chemical) and 10 μL of tetramethylenediamine (TEMED; an accelerator, FUIJFILM Wako Pure Chemical) at 4 °C for more than 8 h, followed by thermal copolymerization at 70 °C for more than 8 h. The thermal method in preference to irradiation was chosen because the PMNA was not transparent, as can be seen in Fig. 2a. The PMN modified with a positively charged hydrogels (Supplementary Fig. 5) was prepared using the same polymerization condition as PAMPS but using 1.5 M (3-acrylamidopropyl) trimethylammonium chloride (APTAC, Tokyo Chemical Industry Co., Ltd.) instead of AMPS.

**Structural evaluation of PMN by BET measurement and SEM observation.** The mass of a piece of naked PMN was measured in the chamber of a BET instrument (BELSORP 18 PLUS, MicrotracBEL) after dehydration at 1 Pa for 5 h, followed by the programmed injection of nitrogen gas into the chamber, while the total volume of nitrogen gas and the inner pressure were monitored. The specific surface area of PMN was calculated by the following BET equations[47].

$$\frac{P/P_0}{V_a(1 - P/P_0)} = \frac{1}{V_m C} + \left(\frac{C - 1}{V_m C}\right)\frac{P}{P_0} \quad (4)$$

$$S = \alpha \, V_m \quad (5)$$

where $P$ [Pa] is the inner pressure of the chamber, $P_0$ [Pa] is the saturation pressure

of adsorbates (nitrogen), $V_\alpha$ [cm³/g] is the absorbed gas volume per the specific material weight at standard temperature and pressure (STP), $V_m$ [cm³/g] is the monolayer absorbed gas volume, $C$ [–] is BET constant, $S$ [m²/g] is specific surface area, and $\alpha (\approx 4.35)$ [m²/cm³] is the occupied area of nitrogen molecules per specific volume.

The size of particles composing the PMN was estimated from the BET specific surface area and the density. Under the assumption that spherical particles with the same size ($\phi$ $x$ [m]) are attached to one another at points, two equations for the surface area and the volume are described as follows.

$$S = 4\pi \left(\frac{x}{2}\right)^2 \cdot k \qquad (6)$$

$$V = \frac{1}{d} = \frac{4}{3}\pi \left(\frac{x}{2}\right)^3 \cdot k \qquad (7)$$

where $S \approx 7.23$ [m²/g] is the specific surface area, $k$ [–] is the number of particles, $V$ [m³/g] is the specific volume of solid, $d \approx 1.28 \times 10^6$ [g/m³] is the density of solid without pores. By solving the simultaneous Eqs. (6) and (7), the particle size can be estimated as $x \approx 0.65$ µm.

The naked PMNs were dehydrated in an oven at 80 °C, and PAMPS-modified PMNs were freeze-dried in a vacuum freeze dryer. These dried samples were sputtered with gold prior to the observation with SEM (VE 9800, Keyence).

**Test of cytocompatibility.** Cytocompatibility was tested by Live/Dead staining for cultured cells in DMEM media (0.1 g/mL) in which the PAMPS-modified PMN soaked in advance (Supplementary Fig. 4). The DMEM sample was prepared by soaking the PMN sterilized by autoclave (LSX 300, TOMY) for 24 h. Normal human dermal fibroblasts (NHDF, P3, $4.71 \times 10^4$ cells/cm²) were seeded to a 12-well dish and cultured for 24 h at 37 °C and 5% $CO_2$. The cell-containing wells were washed twice with PBS, followed by the treatments with the Live/Dead Staining Kit (Dojindo). Three fluorescence images were randomly taken by a confocal microscopy for each well and analyzed using ImageJ (1.52q) to determine the percentage of live and dead cells in each sample ($N = 3$). Viability of the cells was as high as for the control sample that was cultured with a medium without the presoaking of PMN, indicating the high biocompatibility of the PAMPS-modified PMN.

**Preparation of skin sample.** The preparation of the pig skin samples for experiments in Figs. 3, 4, and 5 is as reported in previous paper[75,76]. Briefly, porcine abdominal skin with epidermis, dermis, and hypodermis with thickness of ~4 mm (Landrace swine, 6-month-old, castrated males, not pigmented, DARD Corp.) was transported with ice at ~0 °C without freezing, stored in a refrigerator at 4 °C, and used within 7 days after extraction (Supplementary Fig. 5).

**pH condition.** We conducted almost all experiments by using pH 7 McIlvaine buffer (182 mM) except for the experiments of the transport of FITC-dextran using pH 6 McIlvaine buffer (163 mM), as summarized in Supplementary Table 2. The salt bridges of silicone rubber tubules were prepared with pH 7.4 Ringer's solution (153 mM). The pH of ISF in epidermis is known to be buffered to 7.35–7.45[77].

**Evaluation of DC resistance.** DC ionic resistance of the PMN chip was measured by using a setup reported previously[45,75]. Briefly, a piece of PMN chip pre-loaded with Ringer's solution was put on a 2.0 wt% agarose sheet (FUJIFILM Wako Pure Chemical) containing the Ringer's solution (Supplementary Table 2) with a thin silicone spacer. Two Ag/AgCl electrodes (KCl sat.) were connected to the back face of the PMN and the agarose, as illustrated in Fig. 2c, via salt bridges of silicone rubber tubules (outer diameter 5 mm, inner diameter 3 mm) filled with 2.0 wt% agarose containing Ringer's solution (ca. 0.3 kΩ·cm)[75]. The Ag/AgCl electrodes were connected to the source-meter unit (Yokogawa GS 820), and the potential difference between the electrodes was measured during application of direct currents (1 µA) to calculate the net DC resistance. The potential drop due to the resistance of the measurement system including the tubular salt bridges and the Ag/AgCl electrodes (ca. 50 kΩ) was subtracted from the net resistance.

The measurements for an intact skin and a PMN-inserted skin of human subjects' arms were conducted by using the similar setup on skins instead of the agarose sheet (Fig. 2c). The PMN chips were sterilized by a UV sterilizer for 30 min in advance. The skins of 6 subjects were disinfected with 70% ethanol followed by 3 min drying. The PMNs were pressed lightly (1.0–2.0 N) by hand (see Supplementary Fig. 10), and the agarose gel of the salt-bridge was just put on the PMN or on skin. All procedures performed in studies involving human participants were in accordance with the standards of Ethics Committee of Graduate School of Engineering, Tohoku University (16A-4) and with the 1964 Helsinki declaration and its later amendments. Before experiments, the purpose of this study was explained to subjects who signed the university institutional approved informed consent.

**Evaluation of water transport.** A chip of PAMPS-modified PGMA ($\phi 18$ mm × 1.0 mm of thickness, 0–1.5 M AMPS) was bound with side-by-side Franz cells with a horizontal capillary (inner diameter 1.6 mm), and constant DC currents (0.25–1.0 mA/cm²) were applied by a source-meter (GS 820, Yokogawa) to

evaluate EOF strength of PMN itself without a pig skin. The side-by-side Franz cells were manufactured at a glass workshop of the Technical Division in School of Engineering, Tohoku University (detailed scale, Supplementary Fig. 11). McIlvaine buffer (pH 7.0) (Supplementary Table 2) was used as the electrolyte solution. Ag/AgCl (AgCl + e → Ag + Cl⁻) and Ag (Ag → Ag⁺ + e) were used for a cathode and an anode, respectively[54]. Therefore, no bubbles were observed from the electrodes during the current application. The generation of Ag⁺ at anode seemed no problem during the water transport experiments. The maximum amount of Ag⁺ generated in the present study was ca. 20 µmol (ca. 2.6 mM) for the electrolysis at 0.5 mA for 60 min, which is only ca. 1.4% of the total ions in the buffer solution (ca. 182 mM). Flow velocity of EOF was calculated from the movement of the water surface in a horizontal capillary. To evaluate EOF strength of an intact skin and a PMN-inserted skin, a pig back skin (Landrace swine, 6-month-old, castrated males, not pigmented, DARD) with subcutaneous fat excised was bound with the Franz cells.

**Evaluation of transports of glucose and dextran.** The transports of glucose and dextran were evaluated by using a side-by-side Franz cell handmade of acrylic plates and silicone sheets (Supplementary Fig. 12). The electrodes for current application were the carbon fabrics (CFs) covered by the cellulose semipermeable membranes to prevent the effects of electrolysis. The solutions of 4 ml were poured into the donor and receiver chambers.

For the experiments of transdermal extraction of glucose, an intact pig skin and the skin with PAMPS-modified PMN (1.5 M AMPS) were bound with the side-by-side Franz cells, and into the donor chamber (dermis side) and the receiver chamber (stratum corneum and PMN side) were poured McIlvaine buffer (pH 7.0) containing 5.6 mM glucose (for donor) and 0.11 mM glucose (for receiver), respectively. The pig skin was pretreated by immersing the dermis in McIlvaine buffer containing 5.6 mM glucose at 4 °C for 12 h. A constant DC current (2.0 mA/cm²) was applied by the source-meter in the direction from the dermis side to the stratum corneum side, and a 40 µL portion was collected from the receiver chamber every 15 min for analysis of the increase in concentration of glucose by an assay kit (GAGO20, Sigma-Aldrich) using the calibration curve for the range of 20–80 µg/mL glucose (Supplementary Fig. 13). The initial glucose concentration in the receiver chamber was set to 0.11 mM (20 µg/mL) by considering the minimum range of the assay kit. When the concentration of glucose in the sampled 40 µL solution was over the range (>80 µg/mL), the sample solution was diluted with a buffer without glucose to adjust the concentration between that range. The sampling of 40 µL solution (ca. 1% of the volume of the receiver chamber) was conducted by replacing with a pure buffer solution of the same volume.

For the study of dextran transport, a chip of PAMPS-modified PGMA ($\phi 18$ mm × 1.0 mm of thickness, 0–1.0 M AMPS) was bound with the Franz cell filled with McIlvaine buffer (pH 6.0), and 0.75 mg/mL FITC-dextran (10,000 Da, pKa ≈ 6.4, Sigma-Aldrich) was added to the donor chamber. By using the electrodes covered by a cellulose film to prevent electrolysis of FITC-dextran, a constant DC current (3.0 mA/cm²) was applied by the source-meter in direction from the donor side to the receiver side, and a 100 µL portion was collected from the receiver chamber every hour for analysis with fluorescence spectrophotometer. The sampling of 100 µL solution (ca. 2.5% of the volume of the receiver chamber) was conducted by replacing with a pure buffer of the same volume.

For the experiment of transdermal dosing of FITC-dextran, an intact pig skin and the skin with PAMPS-modified PMN (0.05 M AMPS) was set to the Franz cells so that the stratum corneum faced the donor chamber. After application of 0.50 mA/cm² in the direction from the donor side to the receiver side for 30 min, the rinsed specimens were frozen with liquid nitrogen and sliced into 40-µm sections using a cryostat (CM 1950, Leica) for cross-sectional analysis with a fluorescence confocal microscope (LSM 700, ZEISS).

**Demonstration of transdermal electroosmosis driven by enzymatic fructose/ $O_2$ battery.** The FDH-modified anode was prepared according to the method described in the previous study[11,65]. Briefly, a 10 mm × 10 mm piece of carbon fabric (CF) was first covered with acid-treated multi-walled CNTs (Baytube, Bayer Material Science) by dropping a 50 µL aliquot of a 10 mg/mL CNT dispersion containing 1.0 wt% Triton X-100 (MP Biomedicals), and then dried. This procedure was repeated twice on each side of CF. The piece of CNT-modified CF was immersed in a 1 mL stirred McIlvaine buffer (pH 5.0) containing 10 mg FDH enzyme (FCD-302, 145 U mg⁻¹ from Gluconobactor, Toyobo) for 4 h at 4 °C. Also, the BOD-modified cathode was prepared according to the previous studies[11,65]. Briefly, the modification of CNTs on CF was repeated four times to get a thicker CNT coating, followed by the modification of 10 mg BOD (2.5 U mg⁻¹, from Myrothecium, Amano Enzyme). Importantly, a 100 µL of ethanol dispersion of 4 mg/mL CNT and 12.5 mg/mL PTFE (PTFE, FUJIFILM Wako Pure Chemical) was further coated by dropping twice to ensure hydrophobicity for efficient reduction of ambient $O_2$. As explained in the recent report[65], four couples of the FDH-anode (1 cm²) and the BOD-cathode (1 cm²) were series-connected to boost the output voltage four times. The enzyme electrodes were connected using hydrophobically treated carbon fabrics (0.55 Ω/sq), and these connectors served as the internal resistor (totally <10 Ω). Pieces of cotton cloth (Bemcot™, 1-mm thick, Asahi Kasei) were used as the reservoir of 0.2 M fructose as well as the ion conductor connecting the anodes and cathodes in the series circuit. The fructose

contained in the anode solution (200 mM, 0.3 ml) is 60 μmol, which corresponds to a current generation of 0.2 mA for 8 h (5.8 °C). The current–voltage characteristics of the battery was evaluated by using a potentiostat (model2323, ALS) and a variable resistor (2200, 1200, 680, 470, 330, 220, 100, 82, 47, 20, 10, 5.6, 2.7, 1.2, 0.75, 0.56, 0.3, 0.16, 0.022 kΩ). Each polarization curve was obtained 1 min after changing the resistor.

The demonstration of the biobattery-driven dosing of FITC-dextran at anode was conducted using the PMN modified with 0.05 M AMPS and the cotton with 0.3 mL McIlvaine buffer (pH 6.0) containing 0.2 M D-fructose and 0.75 mg/mL FITC-dextran, as illustrated in Fig. 5d. We have used McIlvaine buffer for the investigation of an enzymatic biobattery, especially to maintain the activity of enzyme electrodes for a longer time[65,78]. The cross-sectional fluorescence image of the skin at anode was analyzed with ImageJ after a 1 h application of the enzymatically produced EOF around 0.2 mA/cm$^2$. The extraction of glucose at cathode was conducted using the PMN modified with 1.5 M AMPS and the cotton cloth receiver (8 cm$^2$ × 1 mm; circular shape with a diameter of 1.5 cm and 1 mm thickness) soaked with the 0.3 mL of McIlvaine buffer (pH 7.0), as illustrated in Fig. 5e. The pig skin was presoaked in McIlvaine buffer (pH 7.0) containing 5.6 mM glucose at 4 °C for 12 h. After a 1 h application of the biobattery, a 40 μL portion of the solution in the cotton receiver was sampled for glucose analysis using the assay kit (GAGO20, Sigma-Aldrich). When the concentration of glucose in the sampled 40 μL solution was over the range of the calibration curve (Supplementary Fig. 13), the sample solution was diluted with a buffer without glucose to adjust the concentration between the range of analysis (20–80 μg/mL glucose). For the lower concentration of the glucose than 20 μg/mL, the sample solution was added to the same volume of 40 μg/mL glucose solution to adjust the concentration between that range. For the experiments to study the longer stability of the battery, the patch was covered by a thin silicone film to minimize the evaporation of the solution in the cottons.

**Reporting summary**. Further information on research design is available in the Nature Research Reporting Summary linked to this article.

## Data availability

Source data for all graphs presented in the paper have been provided in a data source table available at Figshare at https://doi.org/10.6084/m9.figshare.13264922. All other data are available from the corresponding author upon request.

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

## Acknowledgements

This work was partly supported by Tohoku University Frontier Research program (FRiD) and by Grant-in-Aids for Scientific Research A (18H04158) and Challenging Exploratory Research (20K21877) from the Ministry of Education, Culture, Sports, Science and Technology, Japan. We acknowledge Prof. H. Takamura at Tohoku University for BET measurements, Dr. L. Liu at Tohoku University for cytotoxicity testing, and the Biomedical Research Core of Tohoku University Graduate School of Medicine for providing the cryostat.

## Author contributions

S.K., K.S., S.Y., and M.N. conceived the ideas. Y.M. conducted BET surface area measurements and N.K. performed measurement of resistance of microneedles. S.K. and K.S. conducted other experiments. S.K., K.S., and S.Y. analyzed data. S.K., H.A., S.Y., and M.N. wrote and revised the manuscript. All authors have given approval to the final version of the manuscript.

## Competing interests

The authors declare no competing interests.
