## [Peer Review File · Nature Communications]

Reviewers' comments:

Reviewer #1 (Remarks to the Author):

In this manuscript, the authors present 1> an array of solid polymer-based porous microneedles with randomly distributed channels to ensure stable transdermal currents of sufficient magnitude. 2> the porous channel of the needles was modified with a negatively charged hydrogel to generate an electroosmotic flow, and to promote the transdermal transport Owing to the lowered skin resistance and the enhanced efficiency of electroosmosis, 3>a built-in enzymatic biobattery (fructose / O₂ battery) can serve as the power source of a skin patch utilizing electroosmotic transdermal molecular transport.

The suggested fabrication of porous needle was already published in previous paper by by employing poly(glycidylmethacrylate) (PGMA) as the crosslinker of trimethylolpropane trimethacrylate (TRIM) and triethylene glycol dimethacrylate (TEGDMA) and PEG as a porogen.

It is necessary to argue what is mainly novel and new achievements in this work, clearly.

One of the reasons for using the porous MNs are the resistance of stratum corneum is so huge that stable transdermal currents are difficult to be induced, so with MNs' penetration, the transdermal resistance can be significantly lowered. Thus, with MNs containing fluid conduits, transdermal drug delivery or ISF extraction promoted by electroosmotic flow (EOF) can be improved greatly.

Therefore, it is impressive to change the concentration of monomer (AMPS) to optimize the suitable condition to generate EOF based on the small or large molecule to be delivered or extracted.

1> However, according to the toxicity of Poly AMPS, authors could check the cytocompatibility of PAMPS-modified porous MNs as they did for the naked porous MNs (Fig.S1). After checking on sigma aldrich PAMPS is corrosive and can cause harm to skin and eyes. I know it is used in small quantities, but such a device is aimed to be used on a daily basis by patient, use that might become harmful by a repetitive exposition. Besides, the material they used for porous MNs, PGMA, is also kind of harmful to human's health according to safety data sheet. This could be the reason they mentioned PLGA for future research in this article.

2> About the Fig.5(d), it is the cross-sectional fluorescence image after 1h application, if they could indicate which part is stratum corneum, epidermis, and dermis layer, is better to understand. Moreover, if they could provide images after 10 mins, 20 mins...to 1h in supplementary material, readers can understand the process of drug delivery by EOF more clearly.

3> It is very nice to have dual function for the MNs patch, but it is said that "The 40 μ L portion of the solution in the cotton receiver between the PMN and the BOD-modified cathode was analyzed by a glucose assay kit after the 1 h application of the power of the biobattery." I am not sure whether it is bothering for users to use glucose assay kit to detect the glucose concentration after application. Maybe they could combine with some glucose sensor to observe the glucose concentration more quickly.

4> Everything is justified for why and how they do things except why they use such design - pillars topped by a 250 microns microneedle. I understand that the pillar is to help microneedles pierce the skin but 250 microns seems to be very small, it is better to explain how they design their microneedles, or to point out.

5> Finally, if the pores are completely filled with PAMPS hydrogel, can we still consider the array porous ? They use porous microneedles term only so it should be ok. But the extraction seems to happen thanks to permeation and electroosmosis which is happening thanks to a membrane (the PAMPS filled porous microneedle).

Reviewer #2 (Remarks to the Author):

1. Kusama et al. describe the study of electroosmosis on the skin when using a porous microneedle array. The integration with biobattery (biofuel cells) is also demonstrated. The integrated energy system provided the current to supply the iontophoretic system (glucose extraction and drug model delivery). This manuscript reveals an interesting, informative, and useful data for research in transdermal drug delivery and biosensing. However, the statement of novelty and new findings is unclear. (The tables showing comparison with existing technologies can be added in the supporting information.) More supportive studies and critical discussion are needed. The accuracy of scientific terms/writing should be improved. Scientific discussion is also needed to be accurate and clear. Several important issues need to be addressed before considering further. Many revision points are required. After major careful revision, if the authors provide detailed discussion and clarify the points, it is very interesting to reconsider.
2. "penetration/extraction of fluids is known to be promoted by the electroosmotic flow (EOF)..." Please also consider a variety of substances. Some can be induced by direct iontophoresis due to the flowing of the charged species by itself. Therefore, please specify to make the writing more accurate.
3. "the electrical barrier of the stratum corneum (resistance, 0.2 M Ω - 5 M Ω) makes it..." Please consider adding ref.
4. Also, need ref. for "The technical limitations of these transdermal iontophoresis applications arise from the barrier ... difficult to induce stable transdermal currents by using a mild safe voltage."
5. "Needles of a microscale length (usually less than 1 mm) make it possible to pass the stratum corneum without reaching blood vessels and nerves." Additionally, please specify the approximated angle value/geometry of the needle tip.
6. "...swellable or dissolvable polymers 23-28. (Then, followed by) Such swellable needles.." This writing seems that swellable-polymer-based needles and dissolvable-polymer-based needles are in the same group?
7. Page 2. The use of 'On the other hand' may not be appropriate. "*On the other hand*, the transdermal iontophoretic.." Not sure how it shows contrast or in a way that is different from the first thing the authors mentioned?
8. The rationale to use 'negative charge' (rather than positive) should be clearly introduced.
9. It is unclear (in Fig 1) how to cover the built-in enzymatic biobattery. How to avoid the evaporation of solutions from the cotton (e.g., 200 mM fructose in the cotton)?
10. Please discuss the rationale of the geometrical design of needles (e.g., 37 needles; 1 mm interval; 300 μ m height for the support; 450 μ m diameter for the support...). For example, why using this needle density, etc. The simple drawing with geometry should be added in the Supporting Information.
11. Please specify whether 250 μ m length of the needle has been optimized or not.
12. The statistical report can be improved. For example, "6 subjects' arms showed significantly larger and scattered resistance (0.2 - 5 M Ω)", please also keep the individual data points in your Supporting Info. For example, you can plot six small dots as your raw data points. Lines showing mean/standard deviation can be added to this new plot. It is clear to include the individual data points (when carrying out 6 subjects). Also, for "40 - 150 k Ω ".
13. How good of your salt bridges? (e.g., internal resistivity? Cross-sectional area? The resistivity can be also included (not only resistance))
14. Referring to Fig 2c. How to control the pressure used to press the PMN and the salt bridge to touch the agarose gel/arm skin?
15. The silver electrode is dissolved (oxidation) into the solution. This would not affect the study? The solution contains Cl-? Could the change of the skin color be observed?
16. For the study of Fig 2cii, how to prepare the skin? Cleaning the subject's skin?
17. "the holes in the stratum corneum are not stable" Please introduce a brief reason (e.g., regarding physiology, etc.).
18. Fig 2e. Why Subject 2 was studied with a shorter period?
19. Please clarify how to monitor the DC resistance (Fig 2e). Which probes/how to press the probes to contact the skin? It is unclear as you have to insert the PMN and also remove it, and control the time. Is it a real-time measurement?
20. Why not increase AMPS (over 1.5)? Considering the transport of water studies.
21. The scale bar should be added to Fig 3a. In addition, the dimension of the setup should be

described in the Supporting Information.

22. Why using 5.6 mM glucose for the study of Fig 6g. 5.6 mM glucose in the water? Or in electrolytes (if not just water, the authors may wish to describe in the caption as well)?
23. For Fig 3. In the Supporting Info. the picture and thickness of the pig skin should be included.
24. Please also report the freshness of the pig skin, along with the skin preparation.
25. For the studies with glucose and FITC-dextran. What is the volume when sampling for the analysis? How to control and maintain the volume in the experimental setup?
26. The charge status is one of the most crucial factors for the iontophoretic system. If the FITC-dextran drug model has $pK_a \approx 6.4$, how the protonation/deprotonation will be (in pH 6 McIlvaine buffer)? How FITC-dextran molecule displays a neutral charge? Please clarify.
27. Please revise to improve the accuracy of this writing. "(2) permeability for molecules larger than 600 Da." This current form can cover even larger than 10 kDa. Please also modify this related text mentioned in your abstract.
28. "...by mild safe voltage through the lowered net resistance" Please record and report the actual voltage that has been applied to maintain the fixed current.
29. Why choosing poly-glycidyl methacrylate (PGMA), rather than poly (lactic-co-glycolic acid) (PLGA)? Please also consider the swelling, dissolution, etc.
30. Please note the linear dynamic range of glucose detection. How to determine the concentration of extracted glucose in the cotton receiver? Also, using an assay kit (GAGO20, Sigma Aldrich)?
31. Please report the statistical data of Fig 5 (bottom, right, the concentration of glucose extracted).
32. Could the authors please elaborate on the consideration of the "transparency" of PMNA? Why and how this consideration links to microneedle performances?
33. What is the rationale to choose McIlvaine buffer (pH 7.0) and buffer (pH 6.0)?
34. How to coat the fabric-based electrode with PTFE? Only dip-coating with CNTs?
35. Please clarify the rationale to use 200 mM. Why not higher?
36. Please elaborate on the rationale to use this specific value of 5.6 mM glucose when soaking the pig skin.
37. The approach to characterize the polarization curve (Fig 5b) is needed to be described. For example, the authors use a variable resistor; how many minutes for each data point? Scan rate?
38. The shape of the polarization curve and related losses should be discussed. I-V plot seems like a linear line. The V, I, R results obtained from this biobattery are important to control the time-course of the transdermal current and the iontophoresis. You do not use an additional resistor (as a load like 10.1002/adhm.201400457)? How to connect the cells? The internal resistance of the four-cell biofuel cell is only used? Fig 1 is unclear.
39. For 'Conclusions', please specify the advantages over other reports and clearly conclude the new findings.
40. For 'Conclusions', the challenges of this work that the authors have addressed should be emphasized.
41. Please include and discuss more the disadvantages/limitations of this reported approach if any.

Minor

42. Figure 1 should be divided to be 1a, 1b, ...
43. Please indicate the detail of Ringer's solution/McIlvaine buffer (e.g., concentration, ionic strength). This can be added to the Supporting Info. "1x" should not be used; it should be Molar.
44. 0.2 M fructose is prepared in McIlvaine buffer? Which pH?

Reviewer #3 (Remarks to the Author):

In this manuscript, the authors presented a microneedle array-assisted electroosmotic flow (EOF) patch for transdermal delivery of large molecules (>600 Da) or extraction of the interstitial fluids. The authors used a built-in enzymatic biobattery to demonstrate the possible construction of an organic EOF patch. However, the EOF concept, MN for delivery and extraction techniques have already been demonstrated in the field. The experimental data are preliminary and insufficient, the

claimed application of the device was not performed on any in vivo models. Overall, the described work does not qualify for publication on Nature Communications. This manuscript could be more suitable for a specialized journal.

1. Since different concentrations of AMPS could influence EOF strength, and the PMN promotion effect largely depends on the molecular constitution in the interstitial fluid, how to standardize and guarantee the glucose quantification sensitivity between individuals? Could the device distinguish different glucose levels at this stage?
2. The authors mentioned the compression fracture force of the PMN but the data was not presented in any figures.
3. The EOF strength of the PAMP and PAMP+skin control groups should be tested to validate the necessity of the PMN for Fig 3f.
4. Will the PMN assisted EOF process disturbs the bioactivity or function of the large molecular cargo? The use of dextran here could not elicit this issue.
5. For Figure 4d, the authors should add the performance of the PMN-only group.

Author's Response to Reviewer 1

Thanks very much for reviewing our manuscript and providing insightful significant comments. According to your suggestions, we sincerely revised the manuscript with all needed explanations and corrections. The revised parts of the manuscript are marked yellow.

We agree with the reviewer's comments concerning the insufficient description of the novelty and achievements of this work. Here we report a newly developed charge-modified porous microneedle (best quality ever) that significantly lowered the transdermal resistance and enabled the generation of transdermal electroosmotic flow (EOF) (world first) for highly promoted drug delivery and ISF extraction (significant). Even larger molecules can be delivered through the interconnected micropores of the needle (advantage). Namely, we solved three issues for advanced transdermal iontophoresis at once by using charge-modified PMN: (1) lowering the transdermal resistance, (2) transporting of larger molecules and (3) generating a larger EOF. We have emphasized these innovative points in the Abstract (p1 line14-24), Introduction (p3 line1-5) and Discussion (p12 line3-10) of the revised version of the manuscript (the parts written in blue). For these revisions, we referred to the reviewer's thoughtful criticisms and advice.

This world's first transdermal EOF device has been realized by utilizing the unique characteristics of the PMN containing charge-modified interconnected micropores. Therefore, we added a table of classification and characteristics of microneedles (Supplementary Table S1). Also, the title of the paper has been changed to the more striking "Transdermal electroosmotic flow generated by a porous microneedle array patch" to emphasize its novelty.

Table S1 Classification and Characteristics of Microneedles (MNs)

Type of MN	Material	Fabrication	Precision in Miniaturization	Mechanical Strength	Transparency	Permeability	Flow Mechanism	Reference Papers
Conventional Solid MN	Metal, Oxide, Polymer	Molding Photolithography	High	High	+			26, 27
Dissolvable MN	Biomaterials	Molding	High	Low	+			28-31
Hydrogel MN	Xerogel (Dried Hydrogel)	Molding	High	Low	+	Need Swelling		32-34
Hollow MN	Metal, Oxide Polymer	Laser Processing Photolithography	Low	High	+	Partly	Pressure Flow	35-37
Surface Porous MN	Metal, Oxide Polymer	Molding	High	High	-			38-40
Porous MN	Metal, Oxide, Polymer	Molding	High	Low	-	Entirely	Pressure Flow	41-45
Charge-Modified Porous MN	Polymer	Molding	High	Low	-	Entirely	EOF	This Study

Gray cells indicate drawbacks.

(1) However, according to the toxicity of Poly AMPS, authors could check the cytocompatibility of PAMPS-modified porous MNs as they did for the naked porous MNs (Fig.S1). After checking on sigma aldrich PAMPS is corrosive and can cause harm to skin and eyes. I know it is used in small quantities, but such a device is aimed to be used on a daily basis by patient, use that might become harmful by a repetitive exposition. Besides, the material they used for porous MNs, PGMA, is also kind of harmful to human's health according to safety data sheet. This could be the reason they mentioned PLGA for future research in this article.

According to the referee's advice, we checked the cytocompatibility of the PAMPS-modified PGMA using dermal fibroblast, and provided explanations in the revised manuscript (p14 line25-27) and Supplemental Fig. S4. Although the monomer AMPS is toxic, the thoroughly washed PAMPS-modified PMN exhibited cytocompatibility, which is enough for the aim of this paper to prove the novel concept of the transdermal generation of larger EOF by the charge-modified PMN. However, as noted by the reviewer, we are now developing a PLGA-made PMN for the future commercialization of a skin patch (p12 line18-22).

Figure S4. Biocompatibility of the PAMPS-modified PGMA PMN. (a) Illustration showing normal human dermal fibroblasts (NHDF) with a density of 4.71×10^4 cells /cm² seeded to a 12-well dish, and cultured for 3 days in two DMEM medias. (b) The Live/Dead staining of the cells cultured with the DMEM medium without presoaking PMN. (c) The Live/Dead staining of the cells cultured with the DMEM medium prepared by presoaking PMN for 24 hours. (d) Cell viability derived using ImageJ for the cases with (+) and without (-) the presoaking of the PMN (N=3).

(2) About the Fig.5(d), it is the cross-sectional fluorescence image after 1h application, if they could indicate which part is stratum corneum, epidermis, and dermis layer, it is better to understand. Moreover, if they could provide images after 10 mins, 20 mins...to 1h in supplementary material, readers can understand the process of drug delivery by EOF more clearly.

According to the referee's advice, we revised Fig 5d by adding labels of the parts of skin. Unfortunately, it was hard to accurately determine the penetration state of molecules in the sliced dermis sample for microscope observation. In order to monitor the progress of penetration, an in-situ observation system with a multiphoton microscope is under development. We accordingly revised the explanation (p10 line16-18) and added the preparation method of skin samples (p16 line28-29) in the revised manuscript.

(3) It is very nice to have dual function for the MNs patch, but it is said that "The 40 μL portion of the solution in the cotton receiver between the PMN and the BOD-modified cathode was analyzed by a glucose assay kit after the 1 h application of the power of the biobattery." I am not sure whether it is bothering for users to use glucose assay kit to detect the glucose concentration after application. Maybe they could combine with some glucose sensor to observe the glucose concentration more quickly.

We thank the reviewer for the insightful comments. A novel technique for sampling ISF was demonstrated in this paper, and the sensing function will be one of attractive topics to be studied next. A few groups, including us, have reported organic patches for sensing glucose and lactate acid in sweat [66-68]. As the reviewer pointed out, the combination of these organic sensors and the present sampling technique will lead to a totally organic patch for analysis of interstitial fluid. We added these discussions to the revised manuscript (p11 line7-9).

(4) Everything is justified for why and how they do things except why they use such design-pillars topped by a 250 microns microneedle. I understand that the pillar is to help microneedles pierce the skin but 250 microns seems to be very small, it is better to explain how they design their microneedles, or to point out.

We appreciate the reviewer's insightful suggestion. We set the length of needle to 250 μm in order to ensure pain-free insertion. The drawback of such a short needle is the difficulty in inserting it into the elastic skin. It has been proven that the cylindrical post structure can support the secure insertion of an even shorter needle [45]. We added the explanation of the significance of the design of the present needle in the revised manuscript (p4 line5-7, line8-12).

(5) Finally, if the pores are completely filled with PAMPS hydrogel, can we still consider the array porous ? They use porous microneedles term only so it should be ok. But the extraction seems to happen thanks to permeation and electroosmosis which is happening thanks to a membrane (the PAMPS filled porous microneedle).

In response to the reviewer, we carefully checked and revised the proper use of the terms of “naked porous microneedle (PMN)”, “hydrogel-filled PMN” and “hydrogel-modified PMN” throughout the revised manuscript.

Author's Response to Reviewer 2

Thanks very much for reviewing our manuscript and providing insightful significant comments. According to your suggestions, we sincerely revised the manuscript with all needed explanations and corrections. The revised parts of the manuscript are marked yellow.

(1) Kusama et al. describe the study of electroosmosis on the skin when using a porous microneedle array. The integration with biobattery (biofuel cells) is also demonstrated. The integrated energy system provided the current to supply the iontophoretic system (glucose extraction and drug model delivery). This manuscript reveals an interesting, informative, and useful data for research in transdermal drug delivery and biosensing. However, the statement of novelty and new findings is unclear. (The tables showing comparison with existing technologies can be added in the supporting information.) More supportive studies and critical discussion are needed. The accuracy of scientific terms/writing should be improved. Scientific discussion is also needed to be accurate and clear. Several important issues need to be addressed before considering further. Many revision points are required. After major careful revision, if the authors provide detailed discussion and clarify the points, it is very interesting to reconsider.

We agree with the reviewer's comments concerning the insufficient description of the novelty and achievements of this work. Here we report a newly developed charge-modified porous microneedle (best quality ever) that significantly lowered the transdermal resistance and enabled the generation of transdermal electroosmotic flow (EOF) (world first) for highly promoted drug delivery and ISF extraction (significant). Even larger molecules can be delivered through the interconnected micropores of the needle (advantage). Namely, we solved three issues for advanced transdermal iontophoresis at once by using charge-modified PMN: (1) lowering the transdermal resistance, (2) transporting of larger molecules and (3) generating a larger EOF. We have emphasized these innovative points in the Abstract (p1 line14-24), Introduction (p3 line1-5) and Discussion (p12 line3-10) of the revised version of the manuscript (the parts written in blue). For these revisions, we referred to the reviewer's thoughtful criticisms and advice.

This world's first transdermal EOF device has been realized by utilizing the unique characteristics of the PMN containing charge-modified interconnected micropores. Therefore, we added a table of classification and characteristics of microneedles (Supplementary Table S1). Also, the title of the paper has been changed to the more striking "Transdermal electroosmotic flow generated by a porous microneedle array patch" to emphasize its novelty.

Table S1 Classification and Characteristics of Microneedles (MNs)

Type of MN	Material	Fabrication	Precision in Miniaturization	Mechanical Strength	Transparency	Permeability	Flow Mechanism	Reference Papers
Conventional Solid MN	Metal, Oxide, Polymer	Molding Photolithography	High	High	+			26, 27
Dissolvable MN	Biomaterials	Molding	High	Low	+			28-31
Hydrogel MN	Xerogel (Dried Hydrogel)	Molding	High	Low	+	Need Swelling		32-34
Hollow MN	Metal, Oxide Polymer	Laser Processing Photolithography	Low	High	+	Partly	Pressure Flow	35-37
Surface Porous MN	Metal, Oxide Polymer	Molding	High	High	-			38-40
Porous MN	Metal, Oxide, Polymer	Molding	High	Low	-	Entirely	Pressure Flow	41-45
Charge-Modified Porous MN	Polymer	Molding	High	Low	-	Entirely	EOF	This Study

Gray cells indicate drawbacks.

(0) "penetration/extraction of fluids is known to be promoted by the electroosmotic flow (EOF)..." Please also consider a variety of substances. Some can be induced by direct iontophoresis due to the flowing of the charged species by itself. Therefore, please specify to make the writing more accurate.

We regret that we have placed too great focus on electroosmotic flow in this paper. As the reviewer pointed out, iontophoretic transport consists of the electrophoresis of charged molecules themselves as well as the electroosmotic flow (EOF) of the solvent (water) that carries ions or neutral species. EOF is generated by the preferential movement of mobile cations (or anions) in the fluid conduits having fixed anions (or cations). It is known that the negative charge of mucopolysaccharides and proteins (e.g., keratin) in epidermal tissue generate EOF. Here, we succeeded to generate larger transdermal EOF by the charge-modified porous needles. We modified the needle by using a negatively charged hydrogel to match the polarity of skin, which is especially effective for the accelerated dosing of neutral and cationic molecules from the anode side to the cathode side. It is possible to select either negative or positive modification, considering the charge state of molecules and the direction of transport (penetration or extraction). We thoroughly revised the explanations of the mechanism of iontophoresis and EOF in the Introduction (p2 line13-18, p6 line15-17) and added Supplementary Fig.S1.

Figure S1. Iontophoresis consists of the electrophoretic motion of the charged molecules themselves as well as the electroosmotic flow (EOF) generated by the fixed charges. (a) Electrophoresis of the molecules (green) without EOF. (b, c) The net motion of the molecules with the EOF generated by the preferential movement of mobile cations, where the smaller (b) and larger (c) amount of negative charges are fixed (gray).

(3) “the electrical barrier of the stratum corneum (resistance, 0.2 MΩ - 5 MΩ) makes it...” Please consider adding ref.

According to the reviewer’s advice, we cited two recent reports [20,21] (p2 line21), and added the explanation that the resistance values in these references are roughly consistent with ours (0.2 MΩ - 5 MΩ) despite the resistance value of intact skin being dependent on the method of measurement, humidity and the measured part of body (p4 line27-29).

(4) Also, need ref. for “The technical limitations of these transdermal iontophoresis applications arise from the barrier ... difficult to induce stable transdermal currents by using a mild safe voltage.”

In addition to the above references [20,21], we cited papers [7,17] indicating that the practical difficulty of iontophoresis is mainly due to the barrier functions of stratum corneum against the DC current and molecular transport (p2 line20-23).

(5) “Needles of a microscale length (usually less than 1 mm) make it possible to pass the stratum corneum without reaching blood vessels and nerves.” Additionally, please specify the approximated angle value/geometry of the needle tip.

We cited the reference [24,25] for the pain-free insertion of microneedles less than 1 mm in length. Also, we added the angle (28°) and geometry of the PMN (cone of \square 0.13mm, 0.25 mm height) (p4 line8), and showed the detailed design in Supplementary Fig. S2.

Figure S2. The detailed geometry of the PMN array chip.

(6) “...swellable or dissolvable polymers 23-28. (Then, followed by) Such swellable needles..” This writing seems that swellable-polymer-based needles and dissolvable-polymer-based needles are in the same group?

The swellable and dissolvable needles are the works of different groups. We revised the sentence (p2 line29).

(7) Page 2. The use of ‘On the other hand’ may not be appropriate. “*On the other hand*, the transdermal iontophoretic..” Not sure how it shows contrast or in a way that is different from the first thing the authors mentioned?

We revised the wording from “On the other hand” to “Also”.

(8) The rationale to use ‘negative charge’ (rather than positive) should be clearly introduced.

We appreciate the reviewer’s insightful suggestion. Electroosmotic flow is generated by the preferential movement of mobile cations (or anions) in the fluid conduits containing fixed anions (or cations). Here we modified the needle by using a negatively charged hydrogel to match the polarity of skin, which is especially effective for the accelerated transport of neutral and cationic molecules from the anode side to cathode side. As the reviewer pointed out, it is possible to select either negative or positive modification, considering the charge state of molecules and the direction of transport (penetration or extraction). We added these explanations (p6 line15-17).

(9) It is unclear (in Fig 1) how to cover the built-in enzymatic biobattery. How to avoid the evaporation of solutions from the cotton (e.g., 200 mM fructose in the cotton)?

We thank the reviewer for the insightful comments. Figure 1b shows photos of the structural overview of the combination of the biobattery and the PMN. For the experiment to study the longer stability of the battery, the patch was covered by a thin silicone film to minimize evaporation of the solution in the cotton. We added these explanations to the revised manuscript (p17 line25-27, caption of Fig.1).

(10) Please discuss the rationale of the geometrical design of needles (e.g., 37 needles; 1 mm interval; 300 μm height for the support; 450 μm diameter for the support...). For example, why using this needle density, etc. The simple drawing with geometry should be added in the Supporting Information.

We set the length of needle to 250 μm in order to ensure pain-free insertion. The drawback of such a short needle is the difficulty in insertion into the elastic skin. The cylindrical post structure can support the secure insertion of an even shorter needle [45]. The needle density was optimized by examining the insertion probability; a higher density of needle decreased the probability because of the lack of local stretching of skin. We added the discussion of the significance of the design of the present needle in the revised manuscript (p4 line5-7, line8-12).

(11) Please specify whether 250 μm length of the needle has been optimized or not.

We appreciate the reviewer’s insightful suggestion. The 250 μm length of needle roughly corresponds to the thickness of the epidermis layer without reaching the sensory nerves in the dermis layer of skin [46]. We added this explanation about the needle length (p4 line5-7).

(12) The statistical report can be improved. For example, “6 subjects’ arms showed significantly larger and scattered resistance (0.2 - 5 $\text{M}\Omega$)”, please also keep the individual data points in your Supporting Info. For example, you can plot six small dots as your raw data points. Lines showing mean/standard deviation can be added to this new plot. It is clear to include the individual data points (when carrying out 6 subjects). Also, for “40 - 150 $\text{k}\Omega$ ”.

We appreciate the reviewer’s insightful suggestion. We added raw data to Fig. 2c.

(13) How good of your salt bridges? (e.g., internal resistivity? Cross-sectional area? The resistivity can be also included (not only resistance))

We explained the characteristics of a salt bridge (p15 line15-16, line24-26), and cited our previous paper using the same system [73].

(14) Referring to Fig 2c. How to control the pressure used to press the PMN and the salt bridge to touch the agarose gel/arm skin?

The PMNs was pressed lightly(1.0~2.0 N)by hand, and the agarose gel of the salt-bridge was just put on the PMN or on skin. The light press of 1.0~2.0 N caused no significant change in the measured resistance as shown in the added Supplementary Fig. S7. We added these explanations (p15 lines22-24).

Figure S7 The trans-skin resistance measured under varying insertion pressure (1.0 ~ 2.0 N) applied to the PMN on human arm skin.

(15) The silver electrode is dissolved (oxidation) into the solution. This would not affect the study? The solution contains Cl-? Could the change of the skin color be observed?

We have used Ag for the anode that works even in McIlvaine buffer (mixture of Na_2HPO_4 and citric acid without Cl) for the transport measurements in the Franz cell [54]. The generation of Ag^+ at the anode (less than $1\mu\text{mol}$) did not affect the measurement; no change of skin color was observed (p15 line35-36).

(16) For the study of Fig 2cii, how to prepare the skin? Cleaning the subject's skin?

We added the pretreatment of the skins of 6 subjects (p15 line22). The measurements were conducted after sterilization with 70% ethanol followed by 3min drying.

(17) "the holes in the stratum corneum are not stable" Please introduce a brief reason (e.g., regarding physiology, etc.).

The holes made by microneedles are not stable after the removal of the needles, because of the physical closure due to the elasticity of skin tissue and physiological wound healing [52,53]. We added these experiment details (p6 line1-2, line5-7).

(18) Fig 2e. Why Subject 2 was studied with a shorter period?

We unified the experiment time for subject 1 and 2 (Fig. 2e).

(19) Please clarify how to monitor the DC resistance (Fig 2e). Which probes/how to press the probes to contact the skin? It is unclear as you have to insert the PMN and also remove it, and control the time. Is it a real-time measurement?

The experiments in Fig.2e were not continuous, but were point measurements. During the period of PMN insertion, the tubular salt-bridge was put on the PMN as in Fig. 2c-iii (caption of Fig.2).

(20) Why not increase AMPS (over 1.5)? Considering the transport of water studies.

Increasing the charge density (higher concentration of AMPS) causes a larger swelling rate during polymerization. Therefore, at concentrations above 1.5M, the polymerization sometimes caused cracks in the needle. We explained this reason for the maximum AMPS concentration of 1.5 M (p8 line2-3).

(21) The scale bar should be added to Fig 3a. In addition, the dimension of the setup should be described in the Supporting Information.

We added a scale bar and the dimensions of the setup to Fig.3a, and added the details of the Franz cell in Supplementary Fig. S8.

Figure S8. The dimensions of the side-by-side Franz cells with a horizontal capillary, manufactured for evaluation of water flow during the DC current application.

(22) Why using 5.6 mM glucose for the study of Fig 6g. 5.6 mM glucose in the water? Or in electrolytes (if not just water, the authors may wish to describe in the caption as well)?

We used a pH7 McIlvaine buffer for the experiments on glucose transport. The concentration of 5.6mM is typical of blood glucose [75]. We added these explanations (caption of Fig.3, p16 line10).

(23) For Fig 3. In the Supporting Info. the picture and thickness of the pig skin should be included. The thickness of the skin sample was ~4 mm (p8 line9) as shown in Supplementary Fig. S5.

Figure S5 The pictures of a piece of pig abdominal skin used for experiments.

(24) Please also report the freshness of the pig skin, along with the skin preparation.

We added information about the skin preparation along with its freshness (p15 line6-9). The extracted skin samples were carried at $\sim 0^{\circ}\text{C}$ without freezing, stored in a refrigerator at 4°C , and used within 7 days after extraction.

(25) For the studies with glucose and FITC-dextran. What is the volume when sampling for the analysis? How to control and maintain the volume in the experimental setup?

The sampling of received solutions of glucose (40 μL) and dextran (100 μL) were conducted by replacing with a buffer of same volume, which were 0.8 % and 2 % of the total volume of the receiver chamber (5 ml), respectively (p16 line14-15, line22-24).

(26) The charge status is one of the most crucial factors for the iontophoretic system. If the FITC-dextran drug model has $\text{pK}_a \approx 6.4$, how the protonation/deprotonation will be (in pH 6 McIlvaine buffer?)? How FITC-dextran molecule displays a neutral charge? Please clarify.

As the reviewer pointed out, the charge status of the transported molecule is important. We thoroughly revised the explanations of the mechanism of EOF (p2 line13-18, p6 line15-17) and added Supplementary Fig.S1. The FITC-dextran molecule has a single negative charge at pH 6 and two negative charges at pH 7 (Xu et al. *Nanoscale Research Letters* 2011, 6:561). The experiments in Fig. 4 were conducted at pH 6 to see clearly the EOF-assisted transport of FITC-dextran. It is worth noting that we observed the transport of FITC-dextran in the same direction also at pH 7. We added these explanations to the revised manuscript (p8 line35 - p9 line1).

(27) Please revise to improve the accuracy of this writing. “(2) permeability for molecules larger than 600 Da.” This current form can cover even larger than 10 kDa. Please also modify this related text mentioned in your abstract.

There is an empirical rule of the maximum molecular size for transdermal penetration (500~600 Da [22,23]). What we wanted to say was that the PMN can overcome this limit of 500 Da-rule, allowing the transport of larger molecules (p2 line21-23, p12 line11-12). To avoid misunderstanding, we removed the concrete number, 600 Da, from the Abstract and Discussion.

(28) “...by mild safe voltage through the lowered net resistance” Please record and report the actual voltage that has been applied to maintain the fixed current.

We deleted the term “mild safe voltage” because the standard of safety is unknown. The voltage values appeared during the measurement of intact skin arm and that with PMN at 1 μA (Fig. 2c) were ca. 1 V and ca. 0.1 V, respectively.

(29) Why choosing poly-glycidyl methacrylate (PGMA), rather than poly (lactic-co-glycolic acid) (PLGA)? Please also consider the swelling, dissolution, etc.

We developed porous microneedles of PGMA for the first time according to the previous report of porous PGMA (J. Courtois, E. Byström and K. Irgum, *Polymer*, 2006, 47, 2603–2611). Unfortunately, the preparation method for porous PGMA using porogen is not applicable for the pre-polymerized PLGA. Therefore, we are developing now a novel process to prepare porous microneedles of PLGA, and will report it in near future (p12 lines20-22).

(30) Please note the linear dynamic range of glucose detection. How to determine the concentration of extracted glucose in the cotton receiver? Also, using an assay kit (GAGO20, Sigma Aldrich)?

We thank the reviewer for the insightful suggestion. A novel technique for sampling ISF was demonstrated in this paper, and the sensing function will be one of attractive topics to be studied next. A few groups, including us, have reported organic patches for sensing glucose and lactate acid in sweat [66-68]. The combination of these organic sensors and the present sampling technique will lead to a totally organic patch for analysis of interstitial fluid. We added these discussions to the revised manuscript (p11 line7-9).

(31) Please report the statistical data of Fig 5 (bottom, right, the concentration of glucose extracted).

We conducted glucose extraction experiments 3 times, and shown the statistical data in the revised Fig. 5e. Also, we added the typical transdermal current (≈ 0) without the PMN to the revised Fig. 5c.

(32) Could the authors please elaborate on the consideration of the “transparency” of PMNA? Why and how this consideration links to microneedle performances?

Our porous needles are opaque due to light scattering (p4 line4-5). In applications for iontophoresis, no problem occurred at all, while the combination with an optical sensor may be troublesome.

(33) What is the rationale to choose McIlvaine buffer (pH 7.0) and buffer (pH 6.0)?

McIlvaine buffer is a conventional, biocompatible buffer composed of disodium phosphate and citric acid. We have used it for the investigation of enzymatic biobattery, especially to maintain the activity of enzyme electrodes for longer time [63,76]. We explained the rationale of the McIlvaine buffer, and cited our previous papers (p17 line18-20). The compositions of buffers we used are now listed in Supplementary Table S2.

(34) How to coat the fabric-based electrode with PTFE? Only dip-coating with CNTs?

The process of dropping 100 μ L of CNT/PTFE dispersion with a pipette and drying was repeated twice (p17 line4-5).

(35) Please clarify the rationale to use 200 mM. Why not higher?

In our previous paper about the fructose / oxygen biobattery, we obtained optimized performance with around 200 mM fructose [63,76]. The fructose contained in the anode solution (200 mM, 0.3 ml) is 60 μ mol, which corresponds to a current generation of 0.2 mA for 8hrs (5.8 C). Also, a higher concentration of substance is known to decrease the activity of the enzyme. We added these explanations to the revised manuscript (p17 line11-12).

(36) Please elaborate on the rationale to use this specific value of 5.6 mM glucose when soaking the pig skin. The concentration of 5.6mM is a typical of blood glucose [75]. We added these explanations (p16 line10).

(37) The approach to characterize the polarization curve (Fig 5b) is needed to be described. For example, the authors use a variable resistor; how many minutes for each data point? Scan rate?

The resistors [k Ω] used were 2200, 1200, 680, 470, 330, 220, 100, 82, 47, 20, 10, 5.6, 2.7, 1.2, 0.75, 0.56, 0.3, 0.16, 0.022. Each set of polarization curve data (Fig 5b) was obtained 1 min after changing the resistors. We added this procedure to the revised manuscript (p17 line14-15).

(38)The shape of the polarization curve and related losses should be discussed. I-V plot seems like a linear line. The V, I, R results obtained from this biobattery are important to control the time-course of the transdermal current and the iontophoresis. You do not use an additional resistor (as a load like 10.1002/adhm.201400457)? How to connect the cells? The internal resistance of the four-cell biofuel cell is only used? Fig 1 is unclear.

The structure and the basic performance of the series connected biobattery was already reported in a recent paper [63]. We added a brief explanation (p17 line8-9), including that the enzyme electrodes were connected using hydrophobically treated carbon fabric (0.55 Ω/sq), and these connectors served as the internal resistor (totally less than 10 Ω).

(39)For ‘Conclusions’, please specify the advantages over other reports and clearly conclude the new findings.

(40)For ‘Conclusions’, the challenges of this work that the authors have addressed should be emphasized.

We appreciate the reviewer’s insightful suggestion. We proved three significant functions of the originally developed ion-conductive porous microneedle (PMN), aiming at applications for iontophoresis: (1) the lowering and stabilizing of skin resistance by low-invasive partial breaking of the stratum corneum, (2) the molecular permeability through the wholly interconnected micropores, and (3) the generation of large electroosmotic flow (EOF) by charge-modification in the micropores. These advantages of the novel ion-conductive PMN are quite unique compared to the existing types of microneedles (Supplementary TableS1), enabling the controlled generation of the transdermal EOF for the first time. Also, the dimensions of the interconnecting pores of ca. 1.0 μm allows transport of larger drugs that break the 500 Da-rule. We added concluding remarks to emphasize the novelty, advantage, and challenge of this work to the revised Discussion section (p12 line3-10)

(41)Please include and discuss more the disadvantages/limitations of this reported approach if any.

We appreciate the reviewer’s insightful suggestion. In order to further increase the EOF strength of the charge-modified PMN, increasing the porosity would be one of the effective approaches. However, as shown in Table S1, the relatively low mechanical strength is a fatal drawback of PMN. We previously studied the relation between the porosity and the mechanical strength of PMN, and found that sufficient mechanical strength for penetrating skin (compression fracture force, ca. 2.5 N) [41] can be ensured by limiting the porosity to less than ca. 50 %. We added this limitation to the revised version (p12 line15-18). Furthermore, we wish to develop a PMN made of the biodegradable material PLGA so that biological problems don’t occur even if the needle breaks in the skin. We added discussions of this necessary improvement of PMN in material biocompatibility (p12 line20-22).

(42)Figure 1 should be divided to be 1a, 1b, ...

Fig1 was divided to 1a and 1b.

(43)Please indicate the detail of Ringer's solution/McIlvaine buffer (e.g., concentration, ionic strength). This can be added to the Supporting Info. “1×” should not be used; it should be Molar.

We deleted “1x” and showed the following compositions of Ringer’s solution and McIlvaine buffer in Supplementary Table S2.

Table S2 Composition of Buffers Used

pH 7.4	Ringer’s solution	147.2 mM NaCl, 4.02 mM KCl, 2.24 mM CaCl ₂
pH 5	McIlvaine buffer	103 mM Na ₂ HPO ₄ , 48.5 mM Citric acid
pH 6	McIlvaine buffer	126.3 mM Na ₂ HPO ₄ , 36.85 mM Citric acid
pH 7	McIlvaine buffer	164.7 mM Na ₂ HPO ₄ , 17.65 mM Citric acid

(44)0.2 M fructose is prepared in McIlvaine buffer? Which pH?

The pH of the fructose solutions were set using McIlvaine buffer at pH 7 for glucose transport and pH6 for FITC-dextran (p17 line17, line23).

Author's Response to Reviewer 3

Thanks very much for reviewing our manuscript and providing insightful significant comments. According to your suggestions, we sincerely revised the manuscript with all needed explanations and corrections. The revised parts of the manuscript are marked yellow.

In this manuscript, the authors presented a microneedle array-assisted electroosmotic flow (EOF) patch for transdermal delivery of large molecules (>600 Da) or extraction of the interstitial fluids. The authors used a built-in enzymatic biobattery to demonstrate the possible construction of an organic EOF patch. However, the EOF concept, MN for delivery and extraction techniques have already been demonstrated in the field. The experimental data are preliminary and insufficient, the claimed application of the device was not performed on any in vivo models. Overall, the described work does not qualify for publication on Nature Communications. This manuscript could be more suitable for a specialized journal.

We agree with the reviewer's comments concerning the insufficient description of the novelty and achievements of this work. Here we report a newly developed charge-modified porous microneedle (best quality ever) that significantly lowered the transdermal resistance and enabled the generation of transdermal electroosmotic flow (EOF) (world first) for highly promoted drug delivery and ISF extraction (significant). Even larger molecules can be delivered through the interconnected micropores of the needle (advantage). Namely, we solved three issues for advanced transdermal iontophoresis at once by using charge-modified PMN: (1) lowering the transdermal resistance, (2) transporting of larger molecules and (3) generating a larger EOF. We have emphasized these innovative points in the Abstract (p1 line14-24), Introduction (p3 line1-5) and Discussion (p12 line3-10) of the revised version of the manuscript (the parts written in blue). For these revisions, we referred to the reviewer's thoughtful criticisms and advice.

This world's first transdermal EOF device has been realized by utilizing the unique characteristics of the PMN containing charge-modified interconnected micropores. Therefore, we added a table of classification and characteristics of microneedles (Supplementary Table S1). Also, the title of the paper has been changed to the more striking "Transdermal electroosmotic flow generated by a porous microneedle array patch" to emphasize its novelty.

Table S1 Classification and Characteristics of Microneedles (MNs)

Type of MN		Material	Fabrication	Precision in Miniaturization	Mechanical Strength	Transparency	Permeability	Flow Mechanism	Reference Papers
Conventional Solid MN		Metal, Oxide, Polymer	Molding Photolithography	High	High	+			26, 27
Dissolvable MN		Biomaterials	Molding	High	Low	+			28-31
Hydrogel MN		Xerogel (Dried Hydrogel)	Molding	High	Low	+	Need Swelling		32-34
Hollow MN		Metal, Oxide Polymer	Laser Processing Photolithography	Low	High	+	Partly	Pressure Flow	35-37
Surface Porous MN		Metal, Oxide Polymer	Molding	High	High	-			38-40
Porous MN		Metal, Oxide, Polymer	Molding	High	Low	-	Entirely	Pressure Flow	41-45
Charge-Modified Porous MN		Polymer	Molding	High	Low	-	Entirely	EOF	This Study

Gray cells indicate drawbacks.

(1) Since different concentrations of AMPS could influence EOF strength, and the PMN promotion effect largely depends on the molecular constitution in the interstitial fluid, how to standardize and guarantee the glucose quantification sensitivity between individuals? Could the device distinguish different glucose levels at this stage?

We appreciate the reviewer’s insightful comments. As the reviewer pointed out, the transdermal extraction efficiency should be different for each molecule, which is a general issue in sampling ISF by reverse iontophoresis [17, 58]. Usually, a prior calibration has been conducted to estimate the exact concentration in ISF [17, 58]. The present sampling technique with PMN can not fully solve this issue and will require a kind of calibration in the analysis of ISF (p8 line25). Importantly, since the transfer efficiency is mainly determined by the barrier property of the stratum corneum for each molecule, the present technique with PMN would be relatively suitable for sampling ISF as it is. For example, the transdermal extraction of glucose was promoted 5 times by the PMN, while the promotion of water flux was only 2 times (Fig. 3), reflecting the higher barrier function of stratum corneum for glucose than water. These results indicate that the glucose concentration in the sampled solution with PMN would be closer to that of ISF. We added these arguments (p8 line21-23).

(2) The authors mentioned the compression fracture force of the PMN but the data was not presented in any figures.

We measured the compression fracture force of PMN to be $2.5 \text{ N} \pm 0.26$ ($N = 8$). As reported in our previous report [44], a metal jig was pushed from above onto the tip of a microneedle at the rate of 4 mm/min. The force on the metal jig and the displacement were simultaneously recorded by a force gauge, and the fracture force was defined as the maximum force before the sharp drop of force. We added these explanations (p13 line23-24) with a typical result of the force measurement (Supplementary Fig. S3).

Figure S3. The measurement of the compression fracture force of a porous needle [44]. A metal jig was pushed from above onto the tip of a microneedle at a rate of 4 mm/min. The force on the metal jig and displacement were simultaneously recorded by a force gauge, and fracture force was defined as the maximum force before the sharp drop of force. A fracture force of $2.5 \text{ N} \pm 0.26$ ($N = 8$) was obtained.

(3) The EOF strength of the PAMP and PAMP+skin control groups should be tested to validate the necessity of the PMN for Fig 3f.

According to the reviewer's insightful advice, we measured the EOF strength for the PAMPS-filled porous PGMA "plate" without microneedles (gray bar, N = 3). The resulting value of EOF strength was between those of the PAMPS-filled PMN + skin (red bar) and the pig skin alone (black bar). The negative charge of mucopolysaccharide in skin tissue is known to serve as the ionic conduit to generate EOF in the experiments with skin alone. The increase of the transdermal EOF (ca. 2 times) was found to be produced by the synergy of the negative charge of PAMPS and the partial breaking of the stratum corneum with the PAMPS-filled PMN. We added these arguments in the revised manuscript (p8 line11-13, line15-17).

(4) Will the PMN assisted EOF process disturbs the bioactivity or function of the large molecular cargo? The use of dextran here could not elicit this issue.

There are many papers of animal experiments, in which iontophoretically dosed molecules (peptide, enzyme, hormone, vaccine etc.) exerted their function in vivo [69-71]. Therefore, it can be assumed that electroosmosis doesn't affect the function of the transported molecules. This time, we determined that the activity of glucose oxidase (GOx) was maintained after the application of the electroosmotic flow. We added these arguments (p12 line28-31) and Supplementary Fig. S6 to the revised manuscript.

Figure S6. The catalytic activity of the glucose oxidase (100 mg/mL Gox, 150 kDa, Toyobo) before (black) and after (red) the application of electroosmotic flow at 0.5 mA/cm² for 42 hours. The catalytic activity was measured by using an absorptiometer (SEC 2020, ALS) and a GOx activity assay kit (MAK097, Sigma Aldrich).

(5) For Figure 4d, the authors should add the performance of the PMN-only group.

According to the reviewer's insightful suggestion, we added the experiment of dextrin injection with a naked PMN to the revised Fig. 4d. The naked PMN itself doesn't transport dextrin (Fig.4a), but the EOF generated by skin tissue could promote the transport of dextrin. However, significant penetration of dextrin has not been observed, although we conducted the experiments 3 times. These results indicate that the PAMPS-modified PMN was necessary to generate a strong enough EOF to effectively promote the dermal penetration of dextrin. We added these arguments in the revised manuscript (p10 line12-13, line14-15).

REVIEWER COMMENTS

Reviewer #1 (Remarks to the Author):

All answers and comments and additional information regarding to reviewers' questions are very clear now based authors revision files.

So, I think this topic is very promising and interesting so, enough to be published.

Reviewer #2 (Remarks to the Author):

1. The authors have addressed and clarified the novelty concern in the revised manuscript. This is an interesting work. However, many scientific points are still needed to be addressed. One key concern is about the use of polymer-based ion-conductive porous microneedles that provide the charged microscopic channel. The study and critical discussion of pH effects are important.
2. Fig S1. Neutral molecules should be also added.
3. "We employed the modification of negatively charged hydrogel according to the polarity of skin. It is possible to select modification of positive charge, considering the charge state of molecules and the direction of transport (penetration or extraction)." The authors should elaborate more. The examples of the cases using positively charged needle versus negatively charged needle should be added. Currently, it is unclear and not so informative.
4. "The generation of Ag⁺ at anode (less than 1 μ mol)" Please add the calculation of the approximation in the Supporting Info. What is the current? Time?
5. Corresponding to Supplementary Fig.S1. The protonation/deprotonation on the FITC-dextran molecule at different pH (lower; equal to; higher than pKa) must be added. The chemical structure of protonated/deprotonated functional groups must be drawn.
6. "during the measurement of intact skin arm and that with PMN at 1 μ A (Fig. 2c) were ca. 1 V and ca. 0.1 V, respectively" Are you sure? The chronopotentiometry was studied? If sure, please report.
7. Referring to the Old Point: "(30) Please note the linear dynamic range of glucose detection. How to determine the concentration of extracted glucose in the cotton receiver? Also, using an assay kit (GAGO20, Sigma Aldrich)?" The authors did not answer the question yet. Your answer is saying which technique you used for glucose detection? It is very confusing now. The reviewer asked about the "glucose analysis".
8. Referring to the Old Comment "(33) What is the rationale to choose McIlvaine buffer (pH 7.0) and buffer (pH 6.0)?" It is still questionable; why sometimes pH 6.0, why sometimes pH 7.0?
9. Which cases you use buffer (pH 6.0)? Please summarize.
10. Which cases you use buffer (pH 7.0)? Please summarize.
11. Please discuss whether your approach is robust and applicable to the real situations. The real skin and fluid can be dynamic; their pH values can be various. The pH in real conditions will affect directly to the charge of the system, strongly affecting the movement of molecules/chemical species. Both extraction and penetration processes will be affected.
12. What is the value of isoelectric point (pI) of poly-glycidyl methacrylate (PGMA) used in this work? Could you please measure it and discuss it, along with the effect?? Materials and molecules (e.g., drug) bearing which charge under the dynamic pH...
13. The authors refer "The concentration of 5.6mM is a typical of blood glucose [75]." However, it is for the skin, and the authors studied the interstitial fluid (ISF), which contains much lower glucose concentration. The authors are claiming that Fig. 5e is related more to blood glucose concentration?
14. "The measurements were conducted after sterilization with 70% ethanol followed by 3min drying." The authors may not call it "sterilization". Please check the definition (e.g., sterilization, disinfection, clean,)
15. Related to the Old Point "(25) For the studies with glucose and FITC-dextran. What is the volume when sampling for the analysis? How to control and maintain the volume in the experimental setup?" and the response of the authors. It is confusing whether the sampling then replacing with a buffer (which is blank) will dilute the concentration?

Reviewer #3 (Remarks to the Author):

Although the authors have tried to address the issues mentioned earlier, I am still not convinced that the novelty of the paper meets Nature Communications standard. They claimed that this charge-modified porous microneedle was the best quality ever, including improving several issues associated with transdermal iontophoresis, such as lowering the transdermal resistance, transporting of larger molecules and generating a larger EOF. These efforts, while encouraging, are associated with optimization level of the existing concepts and techniques. The proposed applications of drug delivery and ISF extraction demonstrated at this stage do not seem promising for in vivo/clinical use. Therefore I do not reverse my original decision. This study is more suitable for a specialized journal.

Author's Response to Reviewer 1:

All answers and comments and additional information regarding to reviewers' questions are very clear now based authors revision files. So, I think this topic is very promising and interesting so, enough to be published.

We thank the reviewer for the careful review of our manuscript and for the positive assessment of our article.

Author's Response to Reviewer 2:

(1) The authors have addressed and clarified the novelty concern in the revised manuscript. This is an interesting work. However, many scientific points are still needed to be addressed. One key concern is about the use of polymer-based ion-conductive porous microneedles that provide the charged microscopic channel. The study and critical discussion of pH effects are important.

We would like to thank the reviewer 2 for the careful and insightful review of our manuscript and for the positive evaluation of our article.

We added a section "pH condition" to Experimental (p15 line21-24) and revised the list of electrolyte solutions used (Supplementary Table S2). We conducted almost experiments by using pH 7 McIlvaine buffer except for the experiments of FITC-dextran (pH 6). The details of the pH effect on the present EOF system will be discussed in the responses to the comments (8) ~ (11).

Table S2 Composition of Electrolyte Solutions Used

	Ringer's solution	147.2 mM NaCl, 4.02 mM KCl, 2.24 mM CaCl ₂	for evaluating DC resistance
pH 5	McIlvaine buffer	103 mM Na ₂ HPO ₄ , 48.5 mM Citric acid	for preparing FDH anode
pH 6	McIlvaine buffer	126.3 mM Na ₂ HPO ₄ , 36.85 mM Citric acid	for experiments of FITC-dextran transport
pH 7	McIlvaine buffer	164.7 mM Na ₂ HPO ₄ , 17.65 mM Citric acid	for all other experiments

(2) Fig S1. Neutral molecules should be also added.

We agree with the reviewer's advice. We added neutral molecules to Fig. S1 in order to complete the explanation of the mechanism of electroosmotic flow.

Figure S1. Iontophoresis consists of the electrophoretic motion of the charged molecules themselves as well as the electroosmotic flow (EOF) generated by the fixed charges. (a) Electrophoresis of the positively charged, neutral and negatively charged molecules (green) without EOF. (b, c) The net motion of the molecules with the EOF generated by the preferential movement of mobile cations, where the smaller (b) and larger (c) amount of negative charges are fixed (gray).

(3) "We employed the modification of negatively charged hydrogel according to the polarity of skin. It is possible to select modification of positive charge, considering the charge state of molecules and the direction of transport (penetration or extraction)." The authors should elaborate more. The examples of the cases using positively charged needle versus negatively charged needle should be added. Currently, it is unclear and not so informative.

We thank the reviewer for the insightful comments. In order to make the explanation of iontophoresis more clearly, we revised the description (caption of Figure 1, p6 line15-17) and showed the preliminary result obtained by the modification of positively charged hydrogel (poly-(3-acrylamidopropyl) trimethylammonium chloride (PAPTAC)) in the Supplementary Figure S5, in which the water flow of opposite direction was observed. The flow rate, which was slower than the case of PAMPS, will be improved by optimization of the polymerization condition of PAPTAC.

Figure S5. The accumulated amount of water transported at 1 mA / cm² through the PAMPS-filled PMN prepared from 1.5 M AMPS (red), the naked PMN (black) and the PAPTAC-filled PMN prepared from 1.5 M (3-acrylamidopropyl) trimethylammonium chloride (APTAC) (blue). pH7 McIlvaine buffer was used. Error bars indicate standard error of mean (N= 3). The preliminary result obtained by the modification of PAPTAC showed the water flow of opposite direction. The flow rate, which was slower than the case of PAMPS, will be improved by optimization of the polymerization condition of PAPTAC.

(4) “The generation of Ag⁺ at anode (less than 1 μmol)” Please add the calculation of the approximation in the Supporting Info. What is the current? Time?

We sincerely thank for the appropriate pointing out about the electrochemical elution of Ag⁺ ion at the anode during the water transport experiments. The amount of Ag⁺ ion of 1 μmol was a rough calculation for a typical condition (0.5 mA/cm² for 5 min); since the area of the connecting hole of the Franz cell was 0.5 cm², the generation of ca. 0.8 μmol of Ag⁺ ions was estimated using the electrolysis current (0.25 mA), electrolysis time (300 sec) and Faraday constant (96500 C/mol). The maximum amount of Ag⁺ generation in the water transport experiments was ca. 20 μmol (1 mA/cm² for 60 min). The 20 μmol Ag⁺ (ca. 2.6 mM) is only ca. 1.4 % of the total ions in the buffer solution (ca. 182 mM), and in fact there seemed no problem in the experiments. We revised the description regarding the Ag⁺ generation in the evaluations of water transport (p16 line 14-17).

(5) Corresponding to Supplementary Fig.S1. The protonation/deprotonation on the FITC-dextran molecule at different pH (lower; equal to; higher than pKa) must be added. The chemical structure of protonated/deprotonated functional groups must be drawn.

We agree with the reviewer’s advice. We added the drawing of the structures of the FITC-dextran in Supplementary Figure S7 (p8 line35-36). Since the dextran has no charge at the pH conditions used, the protonation/deprotonation of the FITC moiety (pKa = 6.4) determines the charged state of the FITC-dextran.

Figure S7. The chemical structure of FITC-dextran molecule with the protonation/deprotonation on the FITC moiety at different pH [64].

(6) “during the measurement of intact skin arm and that with PMN at 1 μA (Fig. 2c) were ca. 1 V and ca. 0.1 V, respectively” Are you sure? The chronopotentiometry was studied? If sure, please report.

As the reviewer said, we conducted a kind of chronopotentiometry, in which the voltage between the two Ag/AgCl electrodes was measured during application of 1 μA direct current. We revised the explanation (p15 line31-35) in order to explain the measurement method clearer. The resulting voltage was used to calculate DC resistance of the skin or skin with PMN. The potential drop due to the resistance of the measurement system (ca. 50 kΩ) was subtracted.

(7) Referring to the Old Point: “(30) Please note the linear dynamic range of glucose detection. How to determine the concentration of extracted glucose in the cotton receiver? Also, using an assay kit (GAGO20, Sigma Aldrich)?”. The authors did not answer the question yet. Your answer is saying which technique you used for glucose detection? It is very confusing now. The reviewer asked about the “glucose analysis”.

We apologize for misunderstanding the reviewer’s question and failing to make appropriate answers, causing confusion. We have to explain more about the method of glucose analysis especially about the method in the biobattery experiments.

We analyzed the extracted glucose by using the enzyme colorimetric glucose assay kit (GAGO20, Sigma Aldrich). According to the protocol of the kit, the concentration of glucose in the range between 20 ~ 80 $\mu\text{g/mL}$ was measured by using a calibration curve. When the concentration of glucose in the sampled 40 μL solution was over the range, the sample solution was diluted with a buffer without glucose to adjust the concentration between that range. For the lower concentration of the glucose than 20 $\mu\text{g/mL}$, the sample solution was added to the same volume of 40 $\mu\text{g/mL}$ glucose solution to adjust the concentration between the range. We added these explanations to the revised version (p16 line34 -p17 line2) and added the calibration curve (Supplementary Figure S13).

Figure S13. The calibration curve used for calculation of the extracted glucose, which was according to the protocol of the commercialized enzyme colorimetric assay kit (GAGO20, Sigma Aldrich).

For the biobattery experiments in Fig. 5e, a piece of cotton cloth ($1.8\text{cm}^2 \times 1\text{mm}$; circular shape with a diameter of 1.5 cm and 1mm thickness) was used as the receiver of extraction experiments (p18 line10-11). The 0.3 mL of buffer solution was pre-soaked in the cotton receiver, and 40 μL portion of the solution in the cotton was sampled after 1 h application of the biobattery. We revised the description about the sampling / analysis from the cotton receiver (p18 line13-19).

(8)(9)(10) Referring to the Old Comment “(33) What is the rationale to choose McIlvaine buffer (pH 7.0) and buffer (pH 6.0)?”. It is still questionable; why sometimes pH 6.0, why sometimes pH 7.0? Which cases you use buffer (pH 6.0)? Please summarize. Which cases you use buffer (pH 7.0)? Please summarize.

We thank the reviewer for the insightful comments. We conducted almost experiments by using pH 7 McIlvaine buffer except for the transports of FITC-dextran (pH 6), as summarized in Supplementary Table S2. The proportion of mono-anionic FITC-dextran is higher than di-anionic FITC-dextran at pH6 (Supplementary Figure S7). Therefore, the EOF-based transport toward the opposite direction of the electrophoresis of anionic species can be comparatively clear. But importantly, even at pH 7, the transport direction was the same as pH6 due to the strong EOF enough to transport even FITC²⁻ against the electrophoresis. We added some explanation (p10 line1-4) and the results of EOF-based transport at pH 7 as the Supplementary Figure S8(a). Since fluorescence efficiency of FITC is different at pH6 and pH7, the calculation of the amount of FITC-dextran using calibration curves (Figure S8(b)) was necessary for comparison.

Figure S8. (a) The time-course of the accumulated amount of FITC-dextran transported to the receiver chamber through the naked PMN (black) and the PAMPS-modified PMN at pH 6 (red) and pH 7 (blue) during application of 3 mA/cm^2 ($N = 3$). (b) The calibration curves used for the calculation of the amount of FITC-dextran in the receiver chamber from the fluorescent intensity at 520 nm.

(11) Please discuss whether your approach is robust and applicable to the real situations. The real skin and fluid can be dynamic; their pH values can be various. The pH in real conditions will affect directly to the charge of the system, strongly affecting the movement of molecules/chemical species. Both extraction and penetration processes will be affected.

We thank the reviewer for valuable comments. We added the information that the pH of ISF in epidermis is buffered to 7.35~7.45 (p15 line24). We cited a reference paper ([77] Proksch, E. pH in nature, human and skin. Journal of Dermatology 45, 1044-1052 (2018)). Since the sulfo groups of PAMPS are completely dissociated in this pH range, the PAMPS-modified PMN would show robustness in the EOF generation in the real situations.

(12) What is the value of isoelectric point (pI) of poly-glycidyl methacrylate (PGMA) used in this work? Could you please measure it and discuss it, along with the effect?? Materials and molecules (e.g., drug) bearing which charge under the dynamic pH.

We thank for important comment. The functional groups of PGMA (glycidyl group and diol group) have no charge at pH conditions studied. Therefore, the naked PGMA does not contribute to generate EOF, as can be seen in Fig.3b and Fig.4a. We added these arguments in the revised version (p6 line21-22).

(13) The authors refer “The concentration of 5.6mM is a typical of blood glucose [75].” However, it is for the skin, and the authors studied the interstitial fluid (ISF), which contains much lower glucose concentration. The authors are claiming that Fig. 5e is related more to blood glucose concentration?

We apologize for making confusion. The concentration of glucose in ISF is known to correlate to that of blood glucose, while the change of concentration has some time lag [58]. We revised the corresponding descriptions (p8 line20-22). What we are claiming in Fig. 5e is the effect of EOF (with biobattery) for extraction of glucose. For Fig.3g, we claimed that the breaking of stratum corneum by PMN would be of advantage for sampling the solution of which glucose concentration is relatively close to the ISF because the barrier function of stratum corneum is higher for glucose than water (p8 line26-27).

(14) “The measurements were conducted after sterilization with 70% ethanol followed by 3min drying.” The authors may not call it “sterilization”. Please check the definition (e.g., sterilization, disinfection, clean, ...)

We thank the reviewer for the appropriate suggestion. We revised the wording to “disinfection” in the revised version (p16 line3).

(15) Related to the Old Point “(25) For the studies with glucose and FITC-dextran. What is the volume when sampling for the analysis? How to control and maintain the volume in the experimental setup?” and the response of the authors. It is confusing whether the sampling then replacing with a buffer (which is blank) will dilute the concentration?

We thank the reviewer for the important comments. The volume of the sampled (replaced) solution of glucose ($40\mu\text{L}$) and dextran ($100\mu\text{L}$) were 1 % and 2.5 % of the total volume of the solution in receiver chamber (4mL), respectively, and thus the dilution effect with the replacement with the blank buffer were not significant. We added these explanations (p17 line3 and line11).

We apologize for confusion probably due to the lack of explanation about the structure of the Franz cell used for the

evaluation of molecular transport of glucose and FITC-dextran (Fig. 3g and Fig.4). We used a Franz cell handmade of acrylic plates and silicone sheets (Supplementary Figure S12). The solutions of 4 ml were poured to the donor and receiver chambers. The electrodes for current application were the carbon fabrics (CFs) covered by the cellulose semipermeable membranes to prevent the effects of electrolysis. We added these explanations to the revised manuscript (p16 line22-26).

Figure S12. The illustrated structure and a photograph of the Franz cell used for evaluation of molecular transport through skins and PMNs (experiments for Fig. 3g and Fig. 4). The parts of the cell were handmade with acrylic plates and silicone sheets. The electrodes for current application were the carbon fabrics (CFs) covered by the cellulose semipermeable membranes to prevent the effects of electrolysis.

Author's Response to Reviewer 3:

Although the authors have tried to address the issues mentioned earlier, I am still not convinced that the novelty of the paper meets Nature Communications standard. They claimed that this charge-modified porous microneedle was the best quality ever, including improving several issues associated with transdermal iontophoresis, such as lowering the transdermal resistance, transporting of larger molecules and generating a larger EOF. These efforts, while encouraging, are associated with optimization level of the existing concepts and techniques. The proposed applications of drug delivery and ISF extraction demonstrated at this stage do not seem promising for in vivo/clinical use. Therefore I do not reverse my original decision. This study is more suitable for a specialized journal.

We thank the reviewer for the careful review of our manuscript and for valuing our achievements in lowering the transdermal resistance, transporting of larger molecules and generating a larger EOF by taking advantage of the porous microneedle. It is sure these achievements are not only optimization of the existing concepts and techniques. Rather, the EOF-based transdermal penetration/extraction of larger molecules via microneedle array is an essentially novel concept that would not exist without the recently realized porous microneedle, as shown in Supplementary Table S1. We would appreciate it if the reviewer will evaluate the significance of this study, which proposed and demonstrated a novel mechanism of transdermal medical treatment.

The description of novelty of this work is left in blue.

REVIEWERS' COMMENTS

Reviewer #2 (Remarks to the Author):

The authors have revised the manuscript considering suggestions and comments. It is good for publication.